# Individuality across environmental context in *Drosophila melanogaster*

**Thomas F Mathejczyk, Cara Knief, Muhammad A Haidar, Florian Freitag, Tydings McClary, Mathias F Wernet*, Gerit A Linneweber***

Institut für Biologie Abteilung Neurobiologie Fachbereich Biologie, Chemie und Pharmazie, Freie Universität Berlin, Berlin, Germany

**\*For correspondence:**
mathias.wernet@fu-berlin.de
(MFW);
gerit.linneweber@fu-berlin.de
(GAL)

**Competing interest:** The authors declare that no competing interests exist.

## eLife Assessment

There is a growing interest in understanding the individuality of animal behaviours. In this **important** article, the authors build and use an impressive array of high throughput phenotyping paradigms to examine the 'stability' (consistency) of behavioural characteristics in a range of contexts and over time. The results show that certain behaviours are individualistic and persist robustly across external stimuli while others are less robust to these changing parameters. The data supporting their findings is extensive and **convincing**. At the same time, the main analyses focus on a selected subset of the many behavioural metrics recorded, so a large fraction of the acquired data remains only lightly explored; by making these additional data available, the authors provide an invaluable resource for future work to apply alternative analytical frameworks and further mine this rich dataset.

**Abstract** Animal behavior is individually variable, and this variability is often consistent over time, a phenomenon called individuality or personality when multiple traits are involved. However, most studies test individuality in only one environment, even though behavior is known to be context-dependent. Analogous to the human 'person-situation debate,' we asked whether and to what extent behavioral individuality persists across changing environmental situations in *Drosophila melanogaster*. Using established and new behavioral assays, we examined three individual traits, namely exploration, attention, and anxiety, across varying environmental contexts, including temperature, visual cues, and arena shape, in both walking and flying flies. We found that individuality is strongly context-dependent, but even under substantial environmental changes, at least one behavioral trait retained individual-specific variation. Different environmental features did not affect individuality equally; instead, they formed a hierarchy in their influence on behavioral consistency. This hierarchy was supported by generalized linear modeling and hierarchical linear mixed-model analysis. Our results show that, as in humans, individuality in flies persists across different situations, although less strongly than across repeated tests in the same context. These findings establish *Drosophila* as a model for dissecting the developmental, neural, and genetic mechanisms underlying consistent individual differences in behavior across variable environments.

## Introduction

Animal species display enormous interindividual differences in behavior. The temporal consistency of these interindividual differences allows for defining these idiosyncrasies as animal individuality, also called animal temperament, behavioral syndrome, or if the entirety of trait differences between individuals is meant, even animal personality (*Sih et al., 2004*; *Dall et al., 2004*; *Gosling, 2008*; *Réale et al., 2007*). Animal individuality has been described for numerous species and behaviors, including: mouse exploration behavior (*Freund et al., 2013*), great tit feeding preferences (*Partridge, 1976*),

octopus threat responses and feeding behavior (*Mather and Anderson, 1993*), pea aphid threat responses (*Schuett et al., 2011*), as well as vinegar fly handedness (*Ayroles et al., 2015*; *Buchanan et al., 2015*), olfaction (*Honegger et al., 2020*), phototaxis (*Kain et al., 2012*), and object orientation (*Linneweber et al., 2020*). The temporal consistency of individuality allows for discriminating it from short-term behavioral changes, like internal state variations. Furthermore, temporal consistency of behavior allows modifications by learning (*Kahsai and Zars, 2011*) and behavior-based natural selection (*Lapiedra et al., 2018*). Historically, the 'nature vs. nurture' debate (i.e. genome vs. environment) has shaped our understanding of the origins of behavioral individuality (*Galton, 1875*). Stochastic developmental processes are a more recently found factor (*Honegger and de Bivort, 2018*; *Linneweber et al., 2020*), providing an additional layer to gene-environmental interactions (*Traynor and Singleton, 2010*). Most experimental studies on animal individuality have focused on a single behavior set in a singular environment, yet commonly used definitions of animal individuality in predominantly field-based disciplines, such as behavioral ecology (*Réale et al., 2007*), and the definition of human personality entail behavioral consistency across situations (*Mischel, 1968*). Similar to humans, animals will never re-encounter the exact same situation under natural conditions, and animal personality can be an important factor in determining the ecological success of an animal and, as a consequence, its entire population (*Takola and Schielzeth, 2022*). It is, therefore, crucial to understand how an animal's personality or individuality in specific traits interacts with variable environments. Despite decades of interest in animal individuality across situations, the experimental evidence remains insufficient, and the results are contradictory, with studies arguing for (*Briffa et al., 2008*; *Herborn et al., 2010*; *Kralj-Fiser et al., 2007*; *Mowles et al., 2012*) and against (*Coleman and Wilson, 1998*; *D'Eath and Burn, 2002*) individuality in animals across situations with many finding mixed results (*Boissy and Bouissou, 1995*; *Spoolder et al., 1996*; *Werkhoven et al., 2021*). Here, we sought to systematically investigate the consistency of individual behavioral differences across situations using laboratory-controlled variable environmental conditions.

There is ample evidence for the consistency of human personality and animal individuality over time (*Costa and McCrae, 1988*; *Weisbuch et al., 2010*). Intuitively, humans attribute consistent personalities to other humans (*Ross, 1977*). The consistency of human behavior across situations, including environmental context, nevertheless remains highly debated, with several studies arguing for (*Costa and McCrae, 1988*; *Fleeson, 2001*) and against it (*Hartshorne and May, 1930*; *Mischel, 1968*). Summarized as the 'person-situation debate,' psychologists have argued for over a century about the relationship between personality and the situational environment, placing this question at the core of personality psychology (*Mischel, 1968*). Animal models for behavioral consistency across situations provide an opportunity to address this question. In our study, the situation is the sum of all environmental features where a specific behavioral task is performed. The work presented here establishes a quantitative animal model to investigate this and related questions about behavioral consistency across environmental situations.

The vinegar fly *Drosophila melanogaster* provides an enormous toolset for studying individuality and its origins across multiple scales, from neurons to circuits to behavior. This comprises a rapidly unfolding connectome (*Dorkenwald et al., 2024*; *Xu et al., 2020*; *Zheng et al., 2018*), an extensive collection of cell type-specific, binary expression systems (*Luan et al., 2006b*; *Pfeiffer et al., 2010*), and a similarly extensive collection of tools for manipulating neuronal function, including neuronal silencers (*Baines et al., 2001*; *Sweeney et al., 1995*), activators (*Luan et al., 2006a*), and thermogenetic (*Hamada et al., 2008*) and optogenetic (*Klapoetke et al., 2014*; *Pulver et al., 2009*) effectors. Hence, the availability of tools and reagents for manipulating small, defined sets of neurons is no longer a major limitation in answering questions about individual behavioral differences. Instead, what has been missing is an integrative approach combining different high-throughput behavioral assays, which would allow for behavioral screening and classification of large quantities of individual flies for several different behavioral parameters across a variety of environmental contexts.

In recent years, mainly low-throughput, single-fly visual assays like the classical two-stripe Buridan assay (*Bülthoff et al., 1982*; *Götz, 1980*), arenas for tethered walking (*Seelig et al., 2010*) or virtual flight (*Mathejczyk and Wernet, 2020*) have been used to study behavioral individuality in vision-based tasks (*Linneweber et al., 2020*; *Mathejczyk and Wernet, 2019*). In Buridan's paradigm, flies walk between two inaccessible visual targets (two vertical high-contrast stripes). The stripes are presented in a homogenously illuminated surrounding with no other visual or non-visual cues. Apart

from work on individuality involving Dorsal Cluster Neurons (DCNs) (*Hassan et al., 2000*), also known as Lobula Columnar Neuron 14 (LC14) (*Otsuna and Ito, 2006*), Buridan's paradigm has been used to show the importance of the central complex for normal walking speed (*Strauss et al., 1992*) and for studying photoreceptor function or development (*Kiral et al., 2020*; *Strauss et al., 2001*), and visual memory (*Neuser et al., 2008*). Other circular arenas have been frequently used to study locomotory activity without visual stimuli (*Branson et al., 2009*; *Robie et al., 2017*; *Valente et al., 2007*).

In contrast to these low-throughput assays, high-throughput assays for studying the behavioral responses of large amounts of animals have been used primarily for genetic screens. Examples include apparatuses to study phototaxis (*Hirsch and Boudreau, 1958*), geotaxis (*Hirsch, 1959*), olfactory learning (*Jiang et al., 2016*; *Quinn et al., 1974*), and circadian rhythms (*Konopka and Benzer, 1971*). The Ethoscope (*Geissmann et al., 2017*) and MARGO (*Werkhoven et al., 2019*) are recent additions to this repertoire that offer several additional high-throughput assays. Except for the *Drosophila* activity monitors (DAM, Trikinetiks), Ethoscope, and MARGO, these group assays cannot distinguish individual behavioral differences. Only the Ethoscope and MARGO provide some visual modules. Hence, progress towards understanding individuality in visually guided behaviors in a high-throughput fashion remains meager, with few exceptions (*Ayroles et al., 2015*; *Buchanan et al., 2015*; *Kain et al., 2012*).

To investigate the influence of the environment on the consistency of visually guided individuality and to systematically quantify individuality across many behavioral traits and their dependency on the environment in walking flies, we built a new multi-parameter assay called Indytrax. Indytrax is a compact and affordable Mathworks MATLAB-based modular assay and tracker for high-throughput individual ethological quantification. The assay includes swappable modules for multiple high-throughput Buridan paradigms, a Y-maze choice assay for studying the decision-making and locomotor handedness (*Buchanan et al., 2015*), and a simple setup to quantify locomotor activity in large groups of flies (*Branson et al., 2009*). We used this tracker to determine the relative contribution of various environmental features (temperature, visual cues, arena shape) to the consistency of individual behavior. Our results show that identically defined visual stimuli (stripe color, width, and length) do not necessarily evoke the same behavior in an individual when presented in different environments. Conversely, we find that specific individual behavioral traits (attention, exploration, and anxiety, categorized via multiple behavioral parameters) remain consistent even in different environments. Finally, using an LED-based virtual flight arena, we show that attention towards a stimulus remains consistent even across different modalities of locomotion (walking versus flying) when using the same Buridan-type visual stimulus. Hence, we find consistent behavioral traits across situations in our fly model, in analogy to what has been demonstrated for humans (*Costa and McCrae, 1988*; *Fleeson, 2001*). Altogether, our results provide the foundation for understanding the complex interaction between individuality and the environment, including the question of why individual behavior is more unpredictable in variable naturalistic conditions, much like the consistency across situations in human subjects (*Mischel, 1968*).

## Results

The goal of our study was to understand to what extent animal individuality is influenced by situational changes in the environment, i.e., how much of an animal's individuality remains after one or more environmental features change. Based on our previous work on animal individuality (*Linneweber et al., 2020*), we started our analysis in the multiparametric two-stripe Buridan's paradigm (*Figures 1* for technical details) (*Colomb et al., 2012*; *Linneweber et al., 2020*). As expected, we measured interindividually variable, yet remarkably consistent behavioral responses, both on the individual (*Figures 1b and 2a*, *Figure 2—figure supplement 2a*) and mean group level (average of all individuals in the same situation, *Figure 2—figure supplement 1a*), when tested in the exact same situation with all environmental features kept the same. This was true for the five representative behavioral parameters (out of 20 parameters total, see *Supplementary file 1* and *Figure 4—figure supplement 3*, the specific five parameters were chosen as they describe important information about the behavior and importantly can be measured across situations), defined in three behavioral traits: exploration (% of time walked, walking speed), attention (vector strength, angular velocity), and anxiety (centrophobicity). The exploration trait is characterized by parameters describing activity. Parameters related to attention reflect heightened responses to visual cues, but unlike commonly used metrics, such

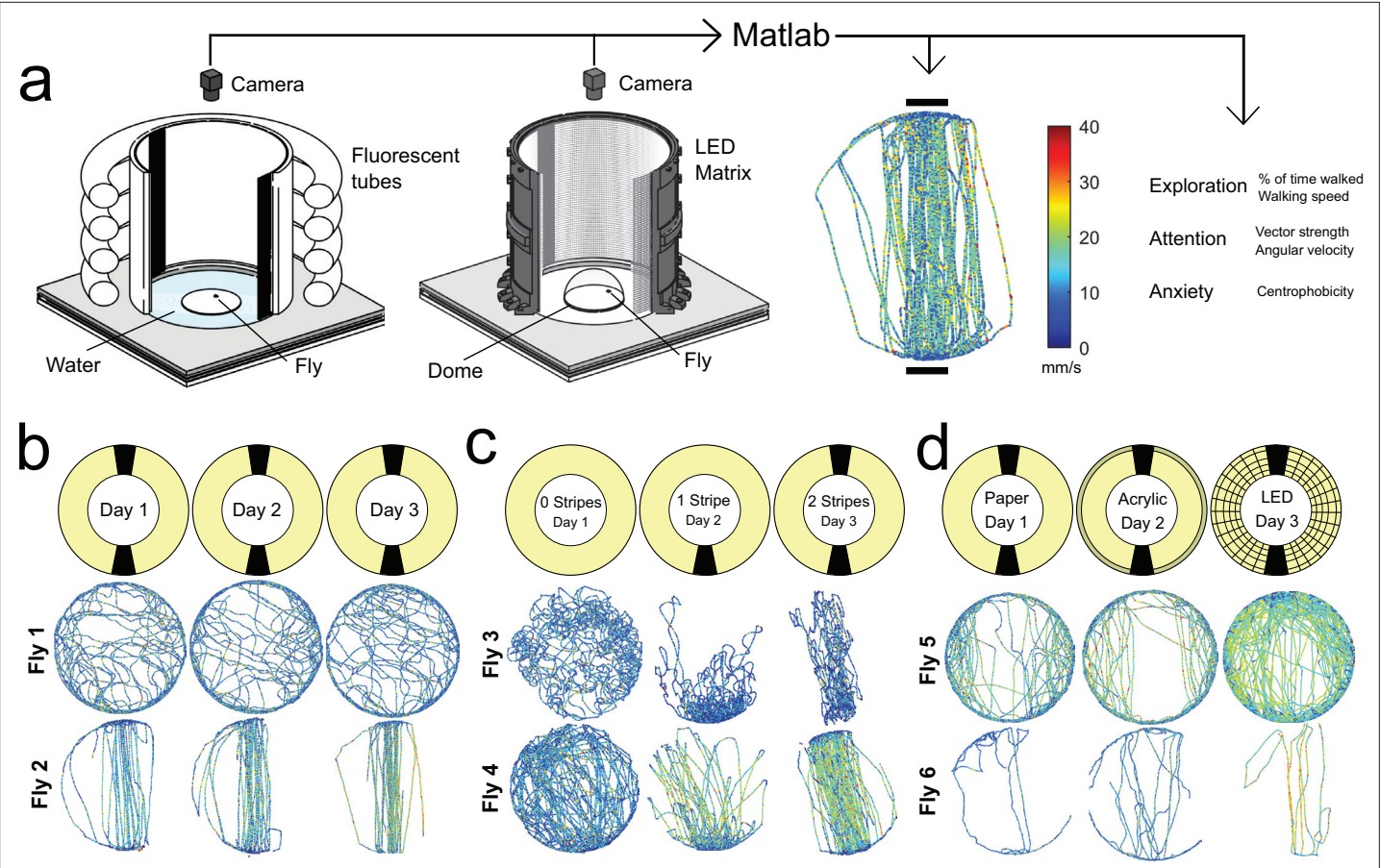

**Figure 1.** Setups for studying individuality in walking flies. (**a**) Overview of a classical (left, fluorescent tubes) and LED (middle) Buridan assay (behavioral platform diameter = 120 mm). A MATLAB script provides single fly tracking and analysis of behavioral key components (right). (**b–d**) persistence of individual walking patterns across time (**b**), different stripe numbers (**c**), and changes in background illumination (**d**).

The online version of this article includes the following figure supplement(s) for figure 1:

**Figure supplement 1.** Construction of Buridan assays.

as angle or stripe deviations (**Colomb et al., 2012**; **Linneweber et al., 2020**), they can be calculated in the absence of visual cues. Lastly, the parameter centrophobicity, used as an indicator of the trait anxiety, is linked to the open-field test in mice, where the ratio of wall-to-open-field activity is frequently calculated as a measurement of anxiety (**Carter and Shieh, 2015**).

Modifying the visual stimulus within the Buridan arena (0, 1, and 2 stripes were tested, **Figures 1c and 2b**, **Figure 2—figure supplements 1b and 2b** and **Figure 2—figure supplement 2b**) significantly affected the mean group behavior across all three behavioral traits (**Figure 2—figure supplement 1b**). Especially, the means of the three parameters (% of time walked, vector strength, and centrophobicity) were strongly affected by the stimulus modification. Surprisingly, despite this change in group means, we observed only a minor effect on individuality when testing the same fly in different situations (**Figures 1c and 2b**, measured by a high Pearson correlation coefficient and a consistent rank within the distribution). The individuality for the traits exploration and anxiety persisted even upon modulation (variation of stripe number) and after removing visual cues from the arena. Only those parameters describing the fly's attention towards visual cues showed an unsurprising reduction in behavioral consistency upon removal of the visual stimuli (**Figure 2b**). Altogether, these results demonstrate that individuality persisted after stimulus modifications within the same arena, even when the modifications resulted in altered group responses.

To further characterize the surprising relationship between individuality and group means upon situational changes, we modified the homogeneity of background illumination in the Buridan arena using two different diffuser materials (paper and acryl-based covers of fluorescent ring lights) and compared

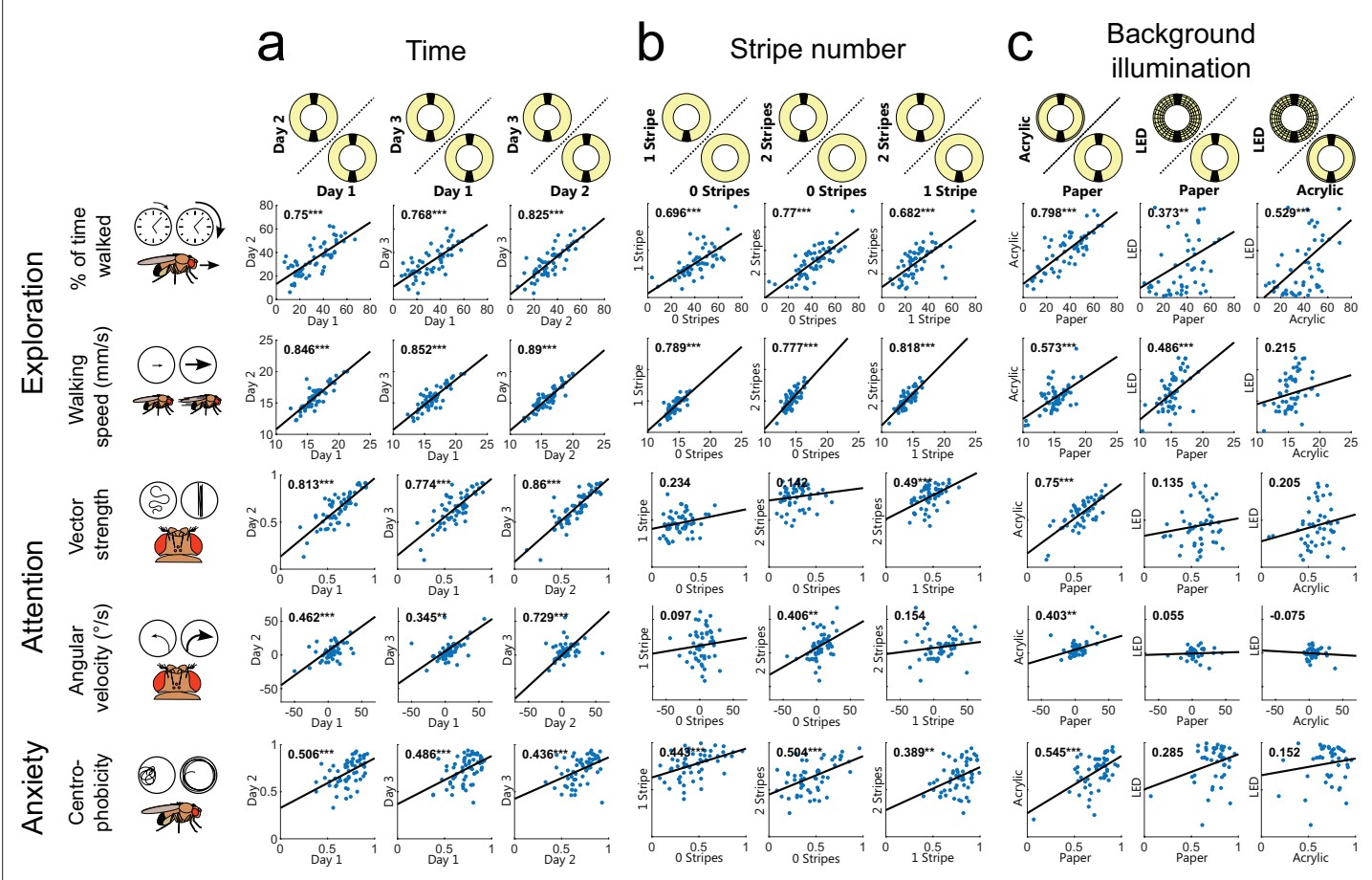

**Figure 2.** Individual traits of walking flies persist over time but depend on the visual scenery. (**a–c**) Correlation of key behavioral parameters over time (n=60) (**a**), different stripe numbers (n=57) (**b**), and changes in background illumination (n=54) (**c**). Correlation coefficients are displayed in the upper left corner of each plot. \*=p<0.05, \*\*=p<0.01, \*\*\*=p<0.001, for detailed fly numbers per experimental condition, see *Figure 2—figure supplement 1*.

The online version of this article includes the following figure supplement(s) for figure 2:

**Figure supplement 1.** Group responses over time, varying stripe numbers, and background illumination.

**Figure supplement 2.** Sex-specific differences in individual traits.

these conditions to a newly designed Buridan LED arena displaying the same visual stimulus (stripes of the same color, width, and height) (*Figures 1d and 2c* and *Figure 2—figure supplements 1c and 2c*). We measured no significant differences between the group means of the two diffuser conditions and only minor group-based changes compared to the LED Buridan (*Figure 2—figure supplement 1c*). We then tested the correlation of individual behavior across these conditions, as a measurement of behavioral consistency, expecting high correlations. Indeed, the different diffusers within the same arenas only had minor effects on the behavior of individual flies (*Figure 1d*). Conversely, the individual response changed dramatically when the same fly was tested again in the LED arena, albeit presenting the same stimulus (*Figure 1d*). A fly's attention and anxiety parameters quantified within the original arena did not allow for any prediction for the LED arena (*Figure 2c*). Significant correlations between the LED arena and the original arenas were only measured for exploration parameters (*Figure 2c*). This unpredictability of individual fly behavior across situations was once again particularly surprising since group-based performances remained similar (*Figure 2—figure supplement 1c*). Finally, a detailed comparison of males and females revealed only quantitative but no qualitative differences across all parameters between the two sexes (*Figure 2—figure supplements 1 and 2*).

Based on the surprising result that individuality (*Gosling, 2008*) of visual orientation behavior is more strongly influenced by the arena type (despite producing very similar group responses) and, only to a lesser degree, dependent on the visual cue (despite strong differences in the group response),

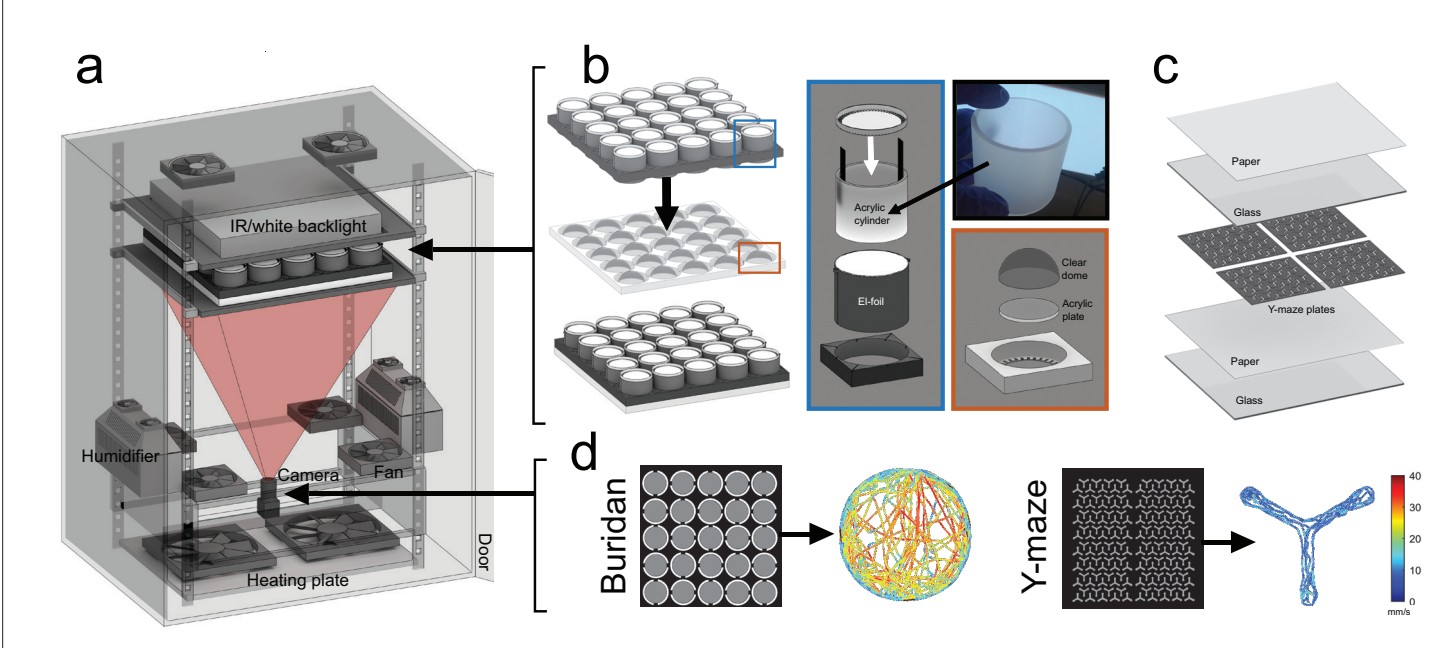

**Figure 3.** A high-throughput assay for studying the influence of temperature, illumination, and arena shape on individuality. (**a**) Setup overview. Flies are filmed from below against a near-infrared backlight and are walking within a 5×5 array of small Buridan chambers (**b**) or one of up to 144 Y-maze arenas (**c**). The setup is housed within a temperature- and humidity-controlled enclosure (600×450 ×820 mm). The provided MATLAB code corrects lens distortions, automatically detects the arenas, and tracks each fly's position over time, allowing for generating high-throughput single-fly behavioral data (**d**). See Materials and methods: Setup 2: IndyTrax multi-arena platform for details.

The online version of this article includes the following figure supplement(s) for figure 3:

**Figure supplement 1.** Construction of a high-throughput assay for studying the influence of temperature, illumination, and arena shape on individuality.

**Figure supplement 2.** Data of a fly group (TopBanana) illustrating orientational responses.

we set out to characterize the contribution of environmental situations to individuality even more quantitatively. We, therefore, built a multiplatform high-throughput assay (Indytrax) to quantitatively study the behavior of the same individual flies in different environmental situations (*Figure 3*, technical details: *Figure 3—figure supplement 1*).

Using Indytrax, we compared the behavioral parameters of each fly in eight different environmental situations (*Figure 4*, *Figure 4—figure supplement 1*, *Figure 4—figure supplement 2*, *Figure 4— figure supplement 3*). The flies were tested in two independent arenas (Buridan and Y-maze) at two different temperatures (23°C and 32°C, *Figure 3—figure supplement 2*) and under both light and dark conditions. Our analysis showed that temperature significantly affected exploration parameters on the group mean level. As expected, most animals tested were more explorative at higher than at lower temperatures (reviewed in: *Gibert et al., 2016*; *Figure 4—figure supplement 1*). Despite these group changes, individuals kept their rank within the group, independent of the changed baseline activity. For example, fly #1 was the least explorative, fly #2 exploration performance ranked medium, and fly #3 the highest (*Figure 4a*). The graphical analysis of exploration parameters (*Figure 4b*) confirmed that most individuals maintain their rank in the distribution under different temperatures and light-dark conditions, whereas the individual distribution rank changed between the different arena types. This relationship was less pronounced for attention parameters. The detailed correlative analysis between individuals underscores these observations (*Figure 4c*, *Figure 4—figure supplement 2* and full analysis *Figure 4—figure supplement 3*), containing also many parameters that can only be measured under specific situations (e.g. deviations from a stripe can only be measured under light conditions and not in the dark): Amongst the exploration parameters, the percentage of time walked significantly correlated between individuals (Pearson $r$=0.263–0.398) independently of temperature, illumination, and arena. Walking speed was highly correlated between individuals (Pearson $r$=0.81, 0.825) under temperature and illumination changes but uncorrelated in different arena types (Pearson $r$=0.091). Vector strength was the most correlated attention parameter. The individual correlations

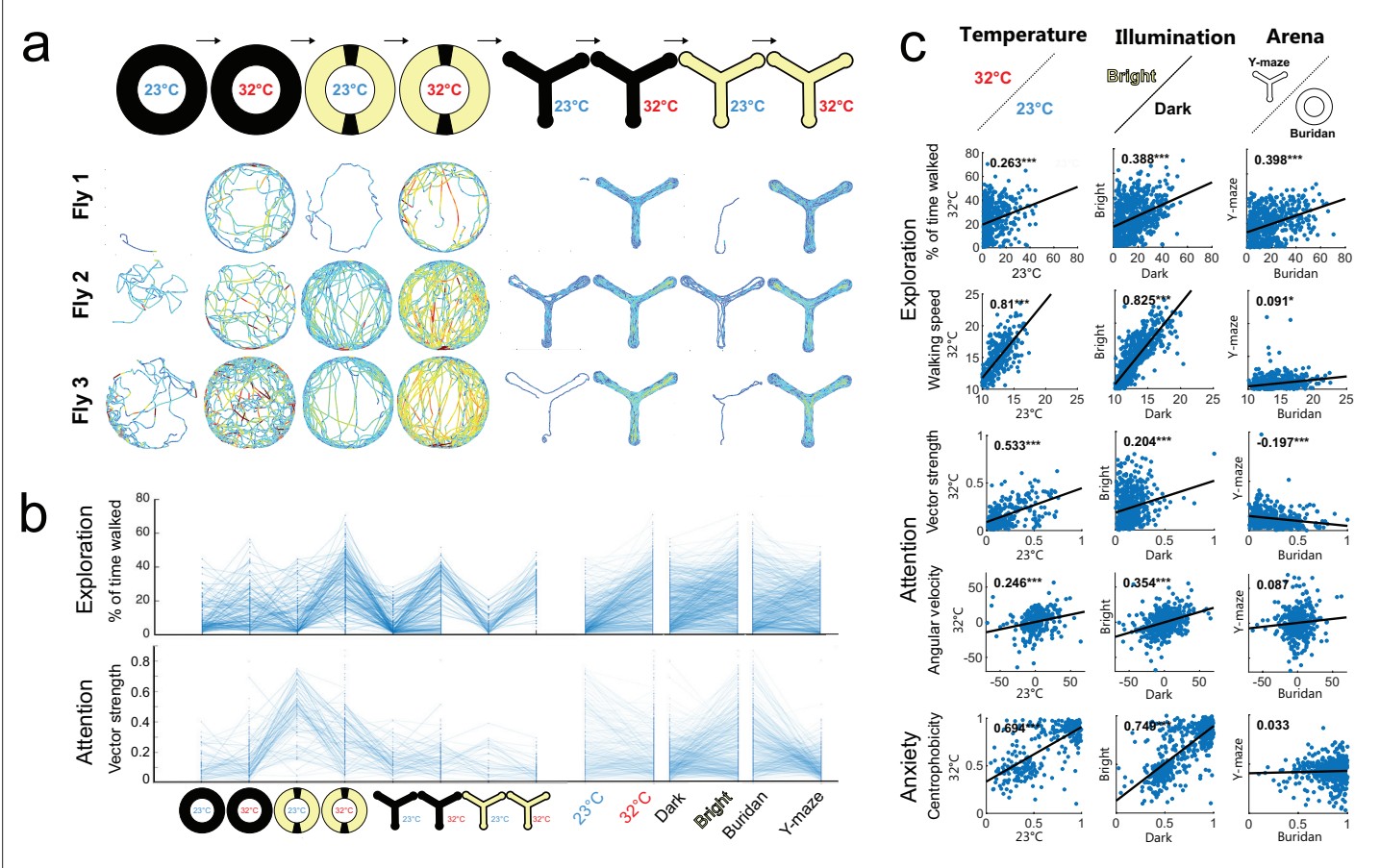

**Figure 4.** The influence of temperature, illumination, and arena shape on individuality in walking flies. (**a**) Individuality in walking patterns across different temperatures, illumination, and arena type. Diameter circular arena = 49 mm, arm length from center Y-maze=13.5 mm. (**b**) Changes in the % of time walked and vector strength for all tested flies depending on environmental conditions (n=400). Right: data grouped by temperature, illumination, and arena. (**c**) Correlation of key behavioral parameters across temperature, illumination, and arena type (n=400). For detailed fly numbers per experimental condition, see *Figure 4—figure supplement 1*.

The online version of this article includes the following figure supplement(s) for figure 4:

**Figure supplement 1.** Sex- and genotype-specific group responses over varying temperature, illumination, and arena shape in walking flies.

**Figure supplement 2.** Sex- and genotype-specific differences based on the influence of temperature, illumination, and arena shape on the persistence of individual traits in walking flies.

**Figure supplement 3.** Complete the correlation matrix across all behavioral parameters and contexts.

ranged from Pearson $r$=0.533 to –0197. Angular velocity significantly correlated with temperature and illumination (Pearson $r$=0.246, 0.354) but not in different arenas (Pearson $r$=0.087). The same is true for anxiety measured through centrophobicity (Pearson $r$=0.694, 0.740, 0.033). In summary, these data revealed that individuality measured in three traits is mostly unaffected by temperature and illumination but strongly affected by the arena type. Our analysis confirmed that even in situations with a large effect on the mean group behavior (like temperature), individual behavioral consistency of idiosyncrasies remained high, while conversely, other environmental situations like the arena type had little impact on the mean but a massive effect on individual behavioral consistency (*Figure 4—figure supplement 1*). Qualitatively similar results leading to the same conclusions were obtained for both sexes (*Figure 4—figure supplement 1a*, *Figure 4—figure supplement 2a*) and two different genotypes (*Figure 4—figure supplement 1b*, *Figure 4—figure supplement 2b*).

Next, we drastically modified the behavioral situation while keeping the visual stimulus as constant as possible. We, therefore, decided to investigate the influence of a fly's modality of locomotion (walking versus flying) on individuality. To do this, we designed a new virtual flight arena for tethered flight utilizing the exact same LED panels as were used in the LED Buridan arena (*Figure 5a-c*,

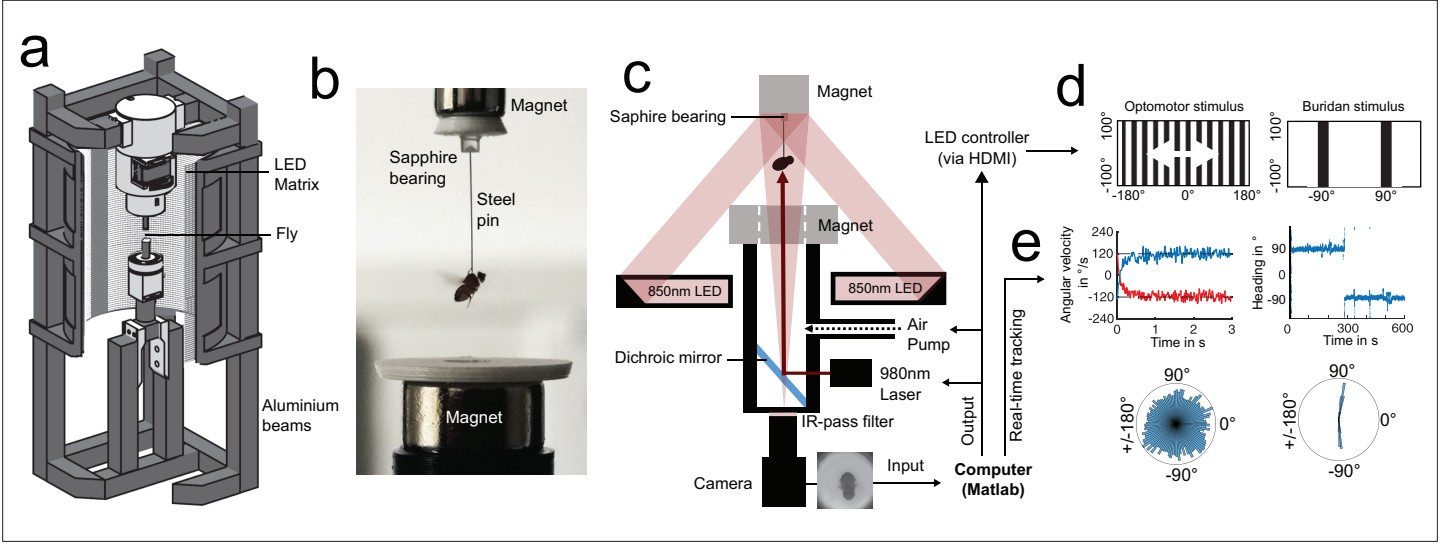

**Figure 5.** A virtual flight simulator for studying individuality in flying flies. (**a**) Setup overview (225×225 ×600 mm). Flies are glued to a steel pin and placed in a magneto-tether surrounded by an LED matrix for stimulus presentation. (**b**) Close-up photo of a tethered flying fly. (**c**) Schematic setup overview. Flies are filmed from below through an IR-pass filter against near-infrared back illumination. A connected pump provides air puffs to reinitiate flight if necessary. We provide a mounting port for an optional IR laser for heat punishment via a dichroic mirror. (**d**) A MATLAB script controls stimulus presentation (optomotor or Buridan stimulus) with an angular extent of 360° azimuthally and 100° vertically and provides simultaneous real-time tracking of flight heading over 360° (**e**). Before an experiment, each fly's ability to rotate was tested using a 120°/s CW or CCW stripe rotation (optomotor stimulus), resulting in measured rotational flight movements close to the rotational velocity of the stripe pattern (**e**), upper left and uniform distribution of angles (**e**), lower left. This was followed by each fly flying under the Buridan stimulus for 10 min (**d**), right and an analysis of heading choices (**e**), upper right and their angular distribution (**e**), lower right.

The online version of this article includes the following figure supplement(s) for figure 5:

**Figure supplement 1.** Construction of a virtual flight simulator for studying the persistence of individual traits in flying flies.

*Figure 5—figure supplement 1* for technical details), to maximize the similarity of the visual context. This flight simulator tracked a fly's heading in real-time in response to dynamic or static panoramic visual stimuli with high resolution (1.4° pixel size). This allowed us to quantitatively compare the visual decisions of both walking and flying flies in response to standard optomotor and Buridan-like stimuli (*Figure 5d–e*). We first tested individuality in flying flies over time under the same and a set of modified visual stimuli (*Figure 6*, *Figure 6—figure supplement 1* and *Figure 6—figure supplement 2*). Each fly was tested over two consecutive days under three different visual conditions (dark stripes on a bright background with either 50% or 100% contrast, and an inverted Buridan with bright stripes on a dark background and 100% contrast) (*Figure 6a*). Overall, the flies showed some individuality in visual attention across days and contrast. This was also true for the inverted Buridan stimulus, in which the flies oriented themselves toward the dark areas, similar to previous results (*Han et al., 2021*). To allow a quantitative comparison across behavioral traits between walking and flight behavior, we derived an adapted set of parameters for exploration and attention quantification (exploration parameters: # of pauses, Absolute angular velocity; attention parameters: Vector strength, Angular velocity, and Median heading axial, *Figure 6b*). The resulting analyses revealed individuality in flying flies (*Figure 6c*). Only the inversion of the Buridan stimulus resulted in variable angles towards the black background on different days. However, it is remarkable that on a given day, each fly selected an arbitrary heading that was kept for the entire experimental length (*Figure 6a*). Again, the quantitative analysis between males and females did not reveal qualitative differences (*Figure 6—figure supplement 2*).

After confirming that individuality also existed in flying flies, we tested how different locomotion modalities affected a given animal's behavioral individuality. Behavioral parameters of both walking and flying flies were quantified in response to the same panoramic LED screen displaying an identical two-stripe Buridan stimulus (*Figure 7a*, *Figure 7—figure supplement 1* and *Figure 7—figure supplement 2*). We expected one of three possible outcomes: (1) There is no consistency of individual behavioral traits across flying and walking. (2) Only some behavioral traits are similar across flying and

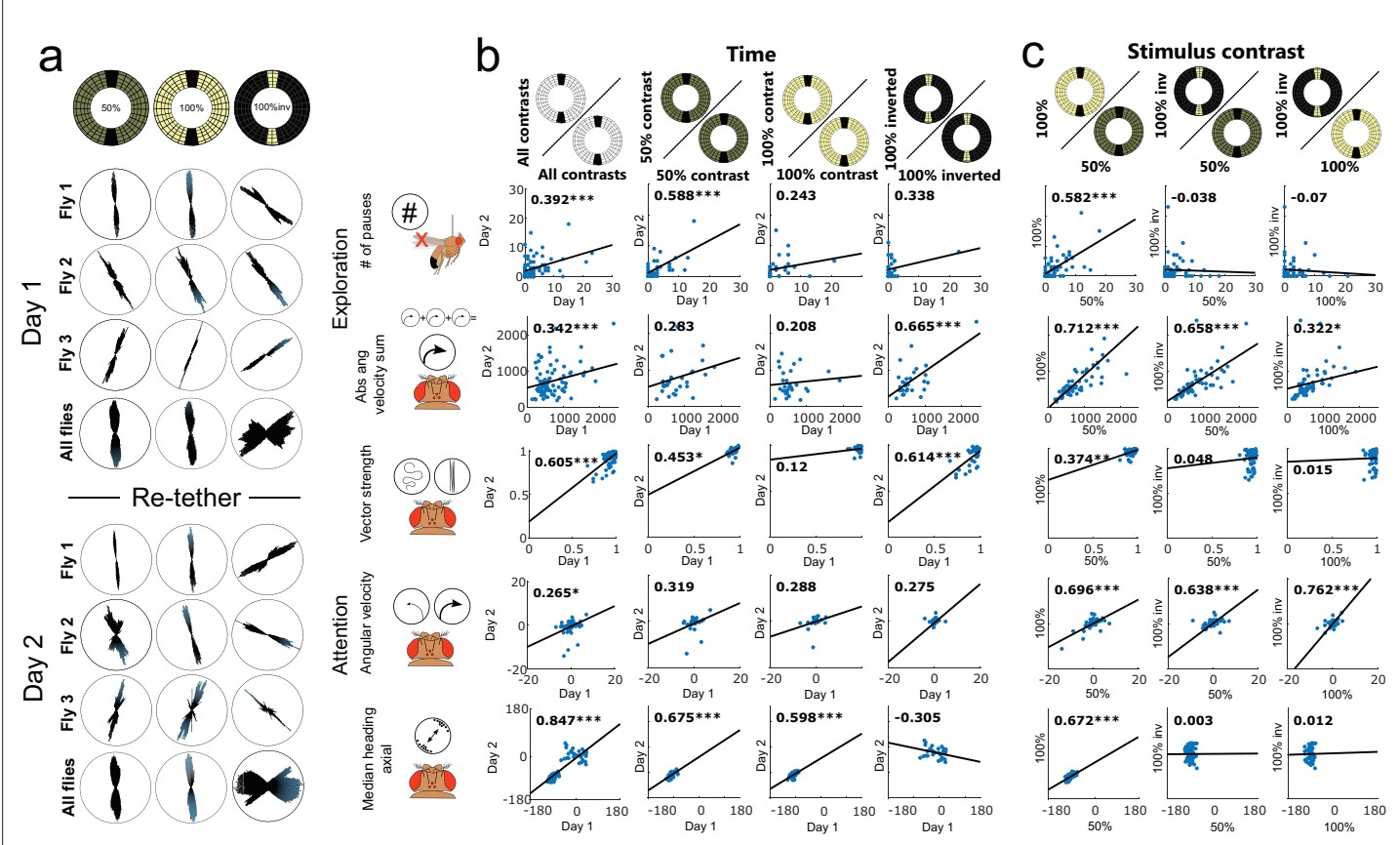

**Figure 6.** Individual traits of flying flies persist over time but depend on the visual scenery. (**a**) Angular histograms show the persistence of individual heading choices over consecutive days and across different contrast ratios. (**b, c**) Correlation of key behavioral parameters over time (n=30) (**b**) and depending on changes in visual contrast (n=30) (**c**).

The online version of this article includes the following figure supplement(s) for figure 6:

**Figure supplement 1.** Group responses over time and different visual contrasts in flying flies.

**Figure supplement 2.** Sex-specific differences in individual traits in flying flies persist over time but depend on the visual scenery.

walking. 3. The fly's individuality remains consistent across the same visual cue, independent of the locomotive modality. Subsequent quantitative analysis indeed revealed a significant correlation for angular velocity between flying and walking flies (**Figure 7b**), indicating that a fly's attention towards a visual object is, to some degree, preserved independent of the locomotor modality. In contrast, we found no statistically significant correlations for any of the exploration parameters between flying and walking. Finally, a more detailed analysis revealed mostly comparable results for male and female data (**Figure 7—figure supplements 1 and 2**).

The data presented here reveal a hierarchical influence of different parameters within the environmental context on individuality (**Figure 8**): First, we conclude that time (at least within the frame of days) has a negligible influence on individual behavioral responses (the basis of the definition of animal individuality in a single situation), since we found consistency of individual behavioral traits and parameters over time in both locomotor modalities, walking and flying. Second, temperature had the second weakest impact on individuality (although only tested for walking behavior). This contrasts with the dramatically altered group mean responses, a difference that can be explained by scaled individual responses in which individuals keep the rank in the group. Third, the nature of the visual input appears to influence the individual orientation responses but has little impact on the exploration parameters. Finally, we identified the most critical situational environmental features that determine individual responses as both the arena type and the locomotor modality (walking vs. flying). Therefore, we conclude that the individuality of visually guided behaviors persists specifically in some

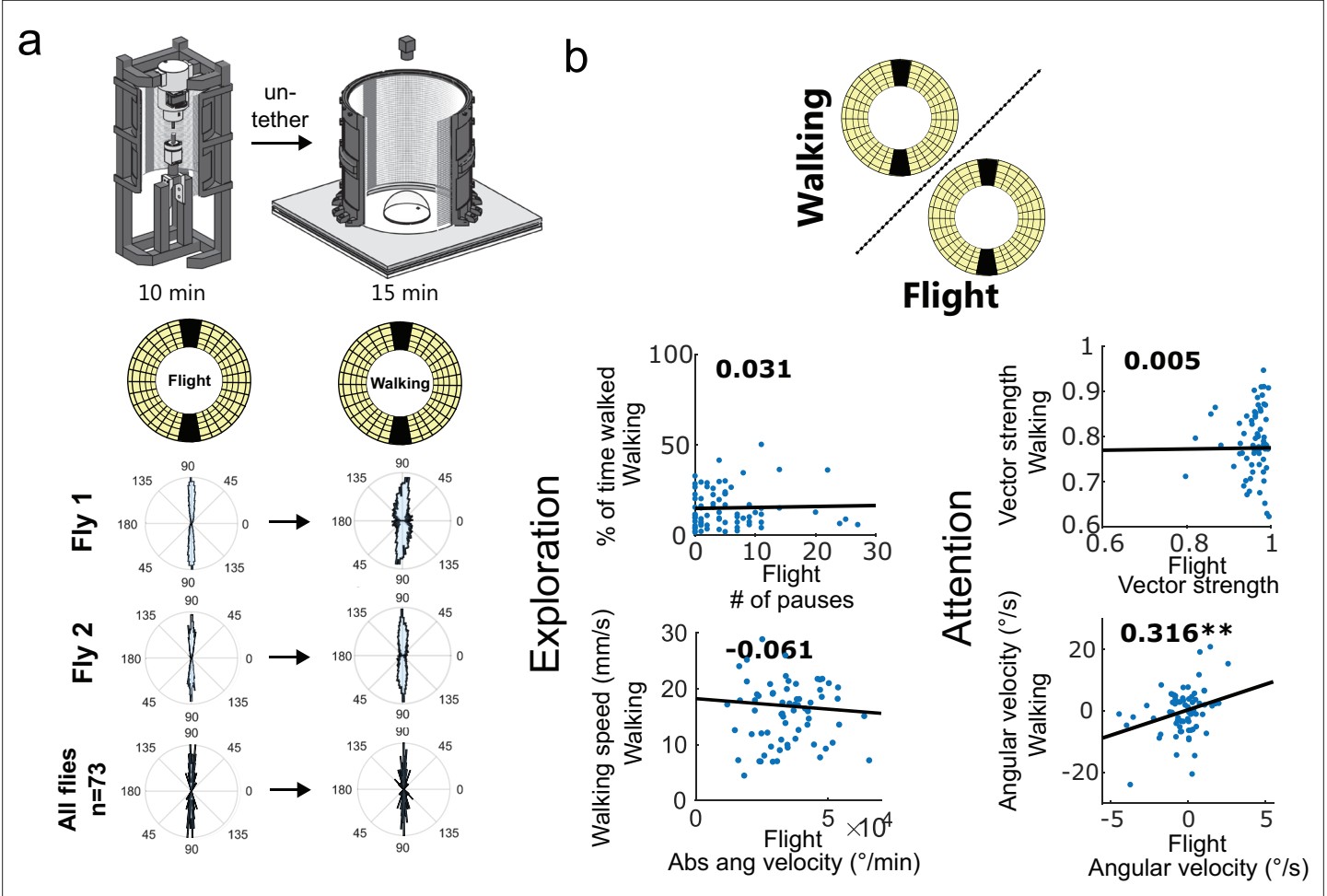

**Figure 7.** Visual attention persists across different behavioral states. (**a**) Experimental overview. Flies were untethered after flying within the flight simulator for 10 min (left) and walking for 15 min (right) under the same Buridan stimulus (stripe size = 11°). Single fly data (middle) and group data (bottom) reveal stripe fixation behavior when walking and during flight. (**b**) Correlation of key behavioral parameters depending on the behavioral state (flight = x axis, walking = y axis, n=73).

The online version of this article includes the following figure supplement(s) for figure 7:

**Figure supplement 1.** Group responses of orientational parameters across different behavioral states.

**Figure supplement 2.** Attention persists across different behavioral states in males but not females.

traits across situations, but the environmental context plays a prominent role in defining individual behavioral responses.

To verify the results of our multiple correlative analyses, we took advantage of a generalized linear model (GLM, *Figure 9*). We first analyzed the effect of time, temperature, vision, and arena on the mean group responses analogous to *Figure 2—figure supplement 1*, *Figure 4—figure supplement 1*, *Figure 6—figure supplement 1*, *Figure 7—figure supplement 1* (*Figure 9a*). Our reanalysis of the walking data confirmed the temperature dependency of activity parameters (*Figure 4—figure supplement 1*). It also revealed effects on the group means by vision and arena, particularly for walking speed. The orientation parameter vector strength was similarly affected by temperature, vision, and arena (see also *Figure 2—figure supplement 1*, *Figure 4—figure supplement 1*), while the position parameter centrophobicity was most strongly affected by the arena (see also *Figure 2—figure supplement 1*, *Figure 4—figure supplement 1*). For flight parameters, the reanalysis of mean data revealed a strong effect of vision on the orientation parameter vector strength and the heading parameter median heading axial (see also *Figure 6—figure supplement 1*). Our reanalysis, therefore, confirmed our previous analysis of mean group data.

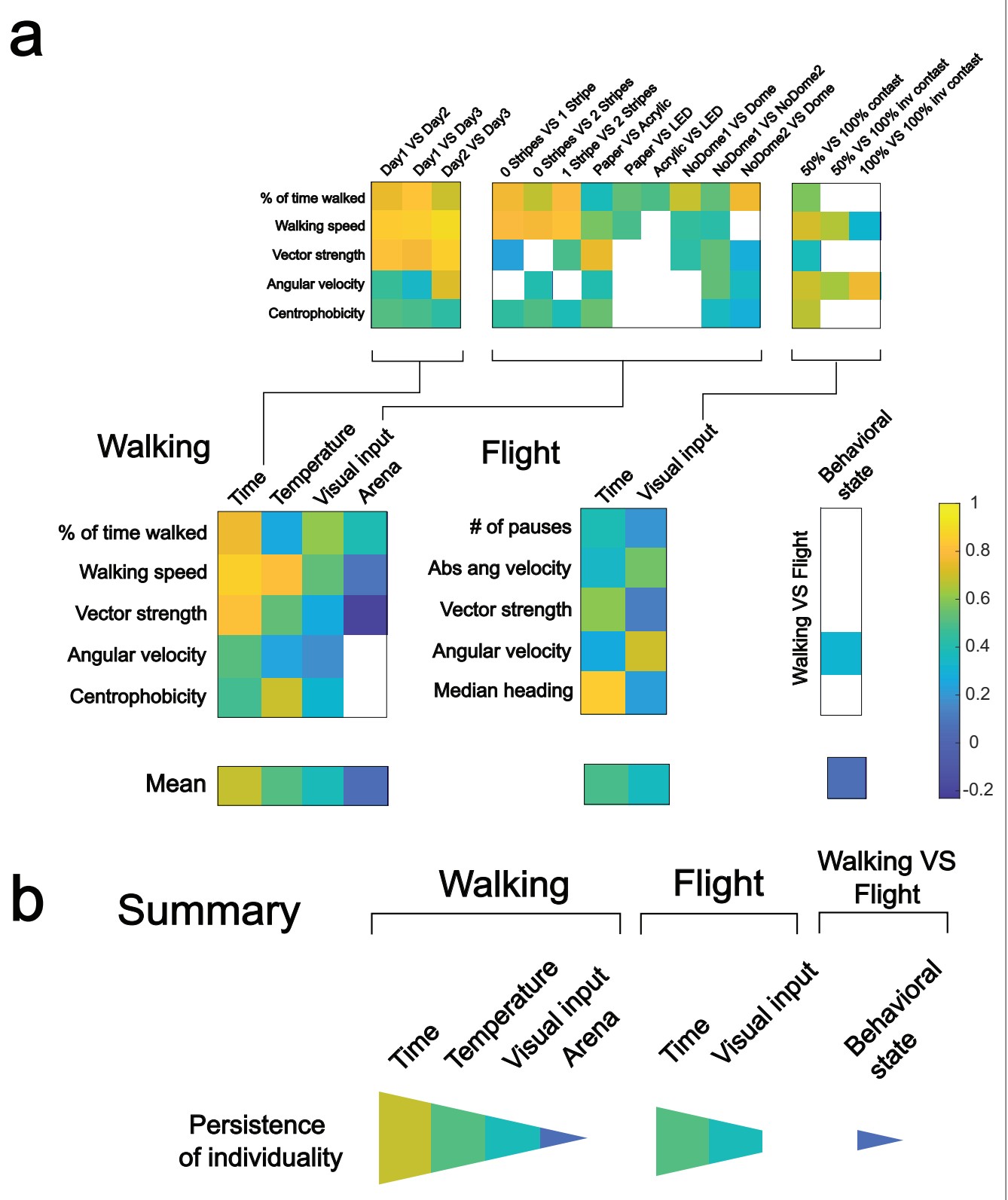

**Figure 8.** A hierarchy of environmental and behavioral contexts based on their influence on individuality. (**a**) Overview summarizing correlation coefficients (*p*<0.05) for key behavioral parameters depending on behavioral context, time, temperature, and visual input. (**b**) Quantification of the influence of behavioral and environmental contexts on the persistence of individuality.

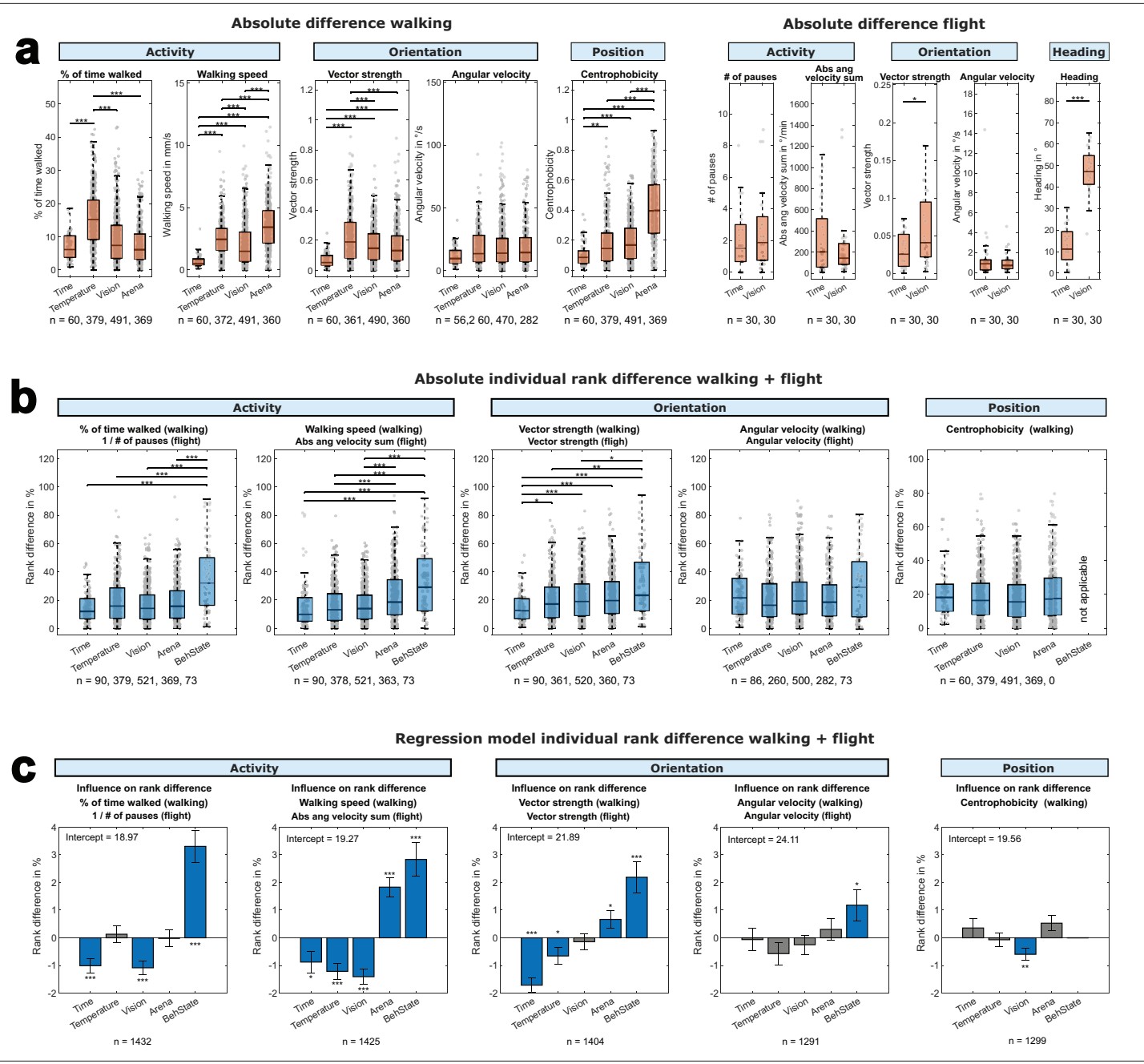

**Figure 9.** A generalized linear model to demonstrate individuality across situations. (**a**) The difference in individual behavioral parameters for walking (left) and flight (right) when changing environmental conditions (time, temperature, vision, arena). Significance: *$p<0.05$, **$p<0.01$, ***$p<0.001$. (**b**) The difference of rank (in %) within the tested groups (walking and flight combined) when changing environmental conditions. Smaller rank differences indicate a lower influence of the environmental parameter on the persistence of individuality. Significance: *$p<0.05$, **$p<0.01$, ***$p<0.001$. (**c**) Generalized linear model (ridge regression) showing the influence of changing environmental parameters on individual rank differences within the tested group (individuality) using the data in (**b**). Significance: *$p<0.05$, **$p<0.01$, ***$p<0.001$. Non-significant environmental parameters are shaded in gray.

The next step of our reanalysis was to compare the rank changes of our flies across situations analogous to the correlation analysis we had done (*Figures 2, 4, 6 and 7*). Our reanalysis revealed significant changes in activity parameters and the orientation parameter vector strength. In these parameters, the effect on the rank of each individual increases from time over temperature, vision, and arena to the behavioral state. Confirming the data we found in *Figures 2, 4, 6–8*. Also, the lack

of a significant effect of the behavioral state for angular velocity confirms our correlation analysis of *Figures 7 and 8*.

Furthermore, we generated a GLM with ridge regression to show the effect of changing environmental situations on individual rank differences (*Figure 9c*). First, we converted the individual mean values for every behavioral trait to within-group ranks (0–100). We then used the GLM to predict the average change in rank for a fly when altering a single environmental context. Each bar in *Figure 9c* represents the predicted influence of changing a certain environmental parameter on the persistence of each behavioral trait, where larger values indicate a larger influence on rank persistence and smaller values indicate less influence of the respective environmental context on rank persistence. Due to the response variable being in rank points, the predicted rank shift due to changing the respective environmental context is the height of a bar in *Figure 9c* added to the indicated intercept values. Based on our correlative analysis, we predicted a spectrum of context dependencies of individual behavioral consistency, where behavioral state > arena > vision > temperature > time in their ability to alter the persistence of individuality. The output of the GLM analysis supports that hypothesis, since calculated coefficients also indicate such a spectrum from rather stabilizing (e.g. time, temperature) to rather disruptive (arena, behavioral state). Our analysis shows that the effect was statistically significant for most environmental parameters.

For the activity parameters, time, temperature, and vision significantly stabilized individual phenotypical rank. In contrast, the arena and behavioral state significantly had a negative effect on rank consistency. For the orientation parameter vector strength, we could fully replicate our previous findings that time had the strongest positive effect on consistency, followed by temperature. Vision had no significant impact. In contrast, the arena had a small negative effect on behavioral consistency, while the behavioral state had a strong negative effect. In the second orientation parameter (angular velocity), a significant destabilizing effect was only found for the behavioral state. For the positional parameter (centrophobicity), only vision had a significant effect on behavioral consistency.

Furthermore, as an additional step in our reanalysis, a hierarchical linear mixed-model analysis confirmed the findings of the previous analyses while providing a more complete and accessible overview of the influence of different environmental contexts on behavioral traits and the individual consistency (repeatability) of these traits across environmental contexts. For walking as well as flight, the majority of behavioral traits showed significant repeatability across different contexts with ICCs on the order of about 0.1–0.6 for walking and 0.2–0.4 for flight (*Figure 10*, see also *Supplementary file 3*). When analyzing walking data, the highest ICCs were measured for activity and orientation-related traits, indicating that flies tend to keep their relative rank for these traits across environmental contexts. In contrast, centrophobicity as a measure for anxiety showed much lower repeatability across different contexts (ICC near 0). The pattern of repeatability was similar for flight traits, with high repeatability for activity traits and low repeatability for the exploratory trait (heading). Both vector strength and the heading showed only low ICCs across different contexts during flight, since the inversion of contrast had a very strong effect on the flies' orientational choices (see *Figure 6*), reducing rank consistency in these traits. Although the flight assays showed slightly lower ICCs overall, possibly reflecting greater trial-to-trial noise, our key finding that specific individual behavioral traits are consistent across different environmental contexts was robust across behavioral modalities (walking vs flight). Group-level behavior was shifted by environmental manipulations (e.g. increasing the temperature led to an increase in mean activity in walking and flight assays). However, individual differences were not erased by these shifts. E.g., in each context, flies that were relatively less active than the average in one environmental context remained so in other environmental contexts, therefore, preserving rank order. Most environmental context effects appeared as shifts in the intercept (group means) rather than changes in the rank order of individual differences.

In summary, using three independent mathematical approaches, we have established a hierarchy of different environmental situations on behavioral consistency. This data demonstrates that temporal stability is more prevalent than consistency across situations in animals. Like humans, animals also have significant consistency across situations, but similar to humans, different situations evoke different behavioral responses, masking some measurable behavioral consistency.

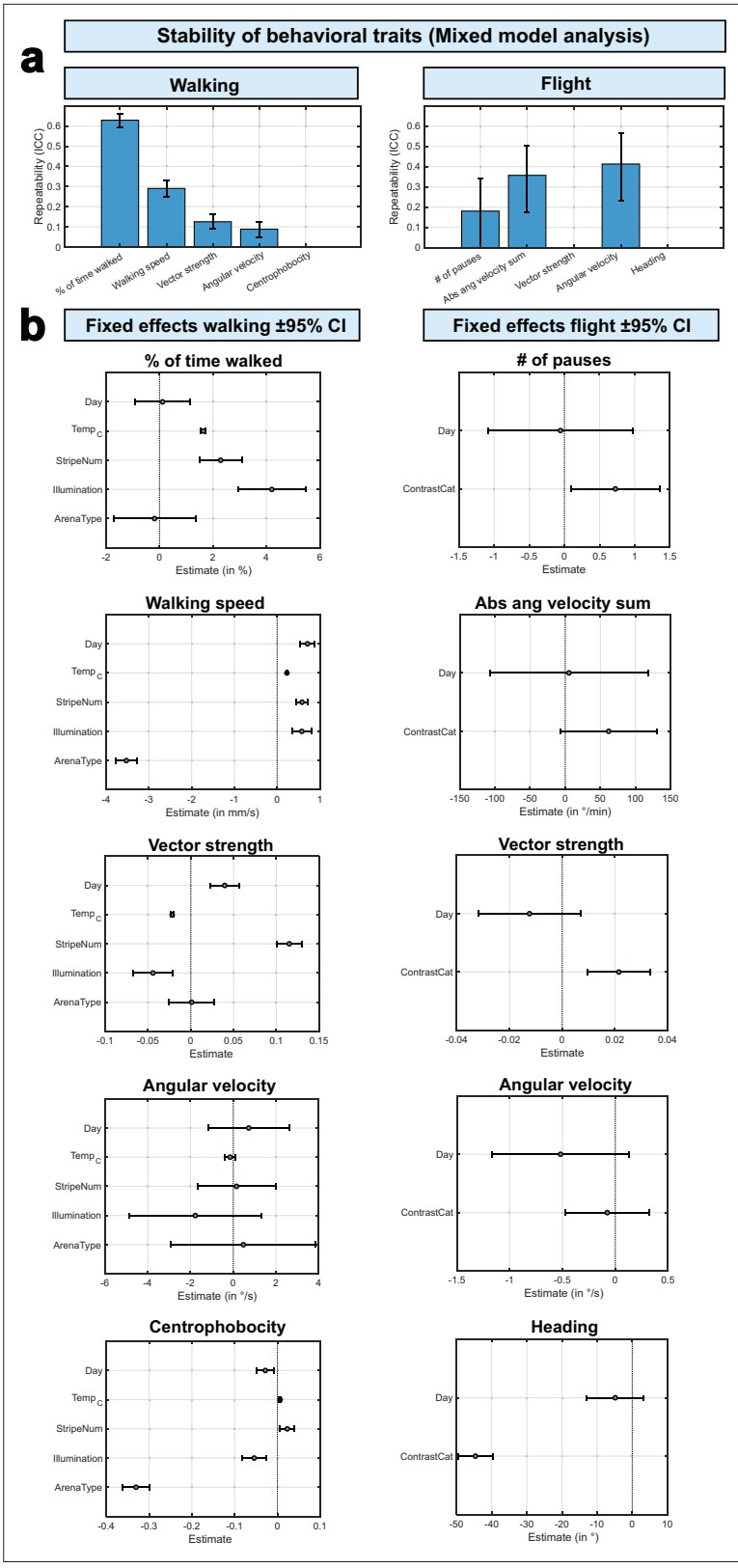

**Figure 10.** Hierarchical linear mixed-model analysis. (**a**) Repeatability across behavioral context of five walking (left) and five flight (right) behavioral traits, as measured via intraclass correlation coefficient (ICC). Bars indicate ICC estimates and whiskers show 95% confidence intervals (CI, obtained via 1000 x bootstrapping, for details, see methods section). High ICC values indicate stronger persistence of individual traits across environmental contexts.

*Figure 10 continued on next page*

*Figure 10 continued*

Number of observations from left to right: 2983, 2912, 2666, 2099, 2987 (walking), 180, 180, 180, 180, 180 (flight). (**b**) Fixed-effect estimates and 95% confidence intervals (whiskers) from hierarchical mixed models for each behavioral trait (left: walking, right: flight). For each modified environmental context (predictors, y-axes), the estimated effects are indicated with points (in units associated with the respective trait) with whiskers representing 95% confidence intervals. Positive estimates indicate that the behavioral trait increases by the indicated values with the predictor increasing one step and vice versa. Predictors were categorized numerically: 1–3 days, 25°–32 °C, 0–2 stripes, light off (0) light on (1), round arena (0) y-maze (1). For a detailed overview of fixed effects, see ***Supplementary file 3***.

## Discussion

Our work presented here focused on the question of whether the combination of consistent behavioral parameters ranging across different traits and, therefore, defining an animal's individuality are invariable across situations or whether they are re-shaped by different environmental contexts. This question is closely related to the questions in the human 'person-situation debate' (***Mischel, 1968***). Until now, in the neurosciences, individuality across situations has been largely ignored since many commonly used definitions of animal individuality only refer to the repeatability of behavior over time in a single environmental situation (***Gosling, 2008***). The contrary is true for mainly field and observation-based sciences, such as ethology and behavioral ecology that have since decades acquired observations for (***Boissy and Bouissou, 1995***; ***Spoolder et al., 1996***) or against (***Coleman and Wilson, 1998***; D'Eath & Burn, 2002) behavioral consistency across situations, with some of the most convincing work probably found in hermit crabs (***Briffa et al., 2008***; ***Mowles et al., 2012***), blue tits (***Herborn et al., 2010***), and geese (***Kralj-Fiser et al., 2007***). Our work establishes animal individuality across situations under several tightly controlled laboratory conditions, revealing that animal individuality shares remarkable similarities to what has been described in human personality research. In agreement with ***Vansteelandt and Van Mechelen, 2004*** our results are in line with the concept of the personality triad, emphasizing the interplay of individual, situation, and behavior (in humans: person, situation, and behavior) (***Vansteelandt and Van Mechelen, 2004***).

Our results confirm that under the same environmental situation and stimulus, animals produce inter-individually variable, but extremely consistent behavioral responses when being retested (***Bell et al., 2009***). Therefore, consistent and individually variable behaviors likely result from individual brain properties that persist over long timescales (in this paper, days as shown in previous papers, months ***Linneweber et al., 2020***), contrasting random behavioral variations like sudden internal state variations. Hence, when presented with the same stimulus in the same situation, each individual responds variably but repetitively, pointing towards a deterministic and hardwired basis of individual properties (***Buchanan et al., 2015***; ***Linneweber et al., 2020***).

The variability of the situational environmental context makes individual animal behavior in the wild much less predictable than that under laboratory conditions (***Briffa and Greenaway, 2011***). In opposition to the vast human literature on the 'person-situation debate' (***Diener and Larsen, 1984***; ***Fleeson, 2001***; ***Mischel, 1968***; ***Vansteelandt and Van Mechelen, 2004***; ***Weisbuch et al., 2010***), the importance of environmental context in shaping individual animal behavior remains descriptive (***Kralj-Fiser et al., 2007***) and experimentally largely unexplored, with few exceptions (***Briffa et al., 2008***; ***Mowles et al., 2012***; ***Versace et al., 2020***; ***Werkhoven et al., 2021***). Here, we closed this gap in knowledge by quantitatively demonstrating how situational environmental and locomotor context changes individual behavior and individuality in animals. Our results demonstrate that situational environmental context has a pronounced effect on individual behavioral responses, be it an alternative stimulus representation, a different arena, or another locomotive modality.

Similar to the human literature (***Mischel, 1968***), the correlation coefficients are lower across situations than in the same situation or environmental context. From this, we can conclude that under more natural conditions with varying contexts, the consistency of individual traits measured by behavioral parameters will be reduced but remain measurable; therefore, for each situational context, the brain computes the appropriate behavior. This is like the situation-behavior profile of the individual in the personality psychology literature (***Mischel and Shoda, 1995***; ***Mischel and Shoda, 1998***). Each time the individual is placed in the same situational context, the same behavior will be evoked; in a different context, a modified behavior. But despite these modifications, the individual brain properties computing the response remain the same. In line with lower but measurable behavioral consistency

across situations, a vast literature demonstrates the importance of animal individual differences for ecological communities under natural conditions (reviewed in: *Sih et al., 2012*; *Takola and Schielzeth, 2022*).

Also, our reanalysis using a hierarchical linear mixed model shows that individuality is consistent but not uniform across behavioral contexts. The repeatability of behavioral traits across environmental contexts was trait-dependent: some behaviors (e.g. edge preference/centrophobicity) exhibited lower intraclass correlation coefficients (ICCs), whereas others (e.g. general locomotor drive and sustained orientation) showed high ICCs. In a multi-context experimental design, such heterogeneity in ICCs is expected and can provide insights into the underlying mechanisms. Behaviors with high ICCs are likely influenced by stable, fly-specific factors (such as developmental or genetic differences), whereas traits with low ICCs may be more sensitive to the immediate environmental context. Importantly, low ICC values do not contradict the existence of individuality; rather, they suggest that inter-individual variation contributes differently to behavior depending on the environmental conditions.

Our quantitative analysis of the effect of environmental and locomotor context on behavioral consistency revealed a hierarchy. As expected from animal and human studies, the time frame of days had little effect on the behavioral outcome (*Bell et al., 2009*; *Buchanan et al., 2015*; *Linneweber et al., 2020*; *Mischel, 1968*; *Pantoja et al., 2016*; *Ross, 1977*; *Schuett et al., 2011*). The hierarchy revealed by our experiments argues for differences in the relevance of environmental features for the traits we measured, which aligns with Mischel's strong or weak situations (*Mischel, 1973*). Therefore, it is unsurprising that exploration can be best measured across situations within experiments of walking flies, and attention towards a stimulus can be measured across movement modalities. Our quantitative analysis demonstrates that situational context can be the most significant factor in determining the individual behavioral outcome, even in a per-design visual assay. This argues that individual behavior is determined by the entirety of the situation, consisting of both the stimulus and the environmental context. For each situation, each individual manifests its individual behavior.

In the future, it will be important to investigate which individual neurophysiological responses change when tested under exactly the same stimulus, but different situations. A systematic analysis of relevant circuit elements for different behaviors will allow for determining the exact neuronal substrates underlying individual behavioral computations within a fly's or other animals' brains. This future research on animal models will provide important new insight into the 'person-situation debate'.

On a more practical level, the importance of the situational environmental context for shaping individual behavioral outcomes is also very important when comparing the results of different research groups when using similar behavioral paradigms. In many instances, it is impossible to replicate all components of a specific assay, let alone the conditions of the experimental room. Our results show that such details substantially impact individual behavioral responses and contribute to errors in replicating the results from another lab (*Crabbe et al., 1999*; *Lithgow et al., 2017*). Only meticulous and detailed reporting of as many environmental features as possible will hopefully alleviate some of these problems in the future. Also, in terms of statistics, using well-established approaches can be an advantage. The hierarchical mixed-model analysis provides a more complete and accessible overview of the findings in our study, making it more comparable to previous findings in behavioral ecology (*Dingemanse and Dochtermann, 2013*).

In conclusion, our work confirms that animals, including flies, have, like humans, individuality across situations. Similar to humans, animal individuality across situations is also marked by lower correlation coefficients than repetitions of a single situation. The individual is consistent across situations, producing individualistic behavior through an interaction with the situation. This involves a hierarchy of influence of different environmental features on the predictability of behavior across situations.

## Materials and methods
### Assay and fly number overview
We tested flies across three main behavioral assays and systematically altered five environmental context parameters, but not all flies could be subjected to all environmental context modifications. In the 'classical' Buridan assay, we tested 60 flies over three days, 57 different flies with three different stripe numbers, and 54 different flies with three different background illuminations. In the high-throughput assay, we tested 400 flies, with each fly tested in two different temperatures, with light

on/off, and in two different arena shapes. In the flight simulator, we tested 30 flies, with each fly tested across two days and three contrast conditions. Furthermore, we tested with 73 different flies in the flight simulator first and then in the 'classical' Buridan assay.

## Animals

All experiments were performed using 1-week-old wild-type *Drosophila melanogaster* (CantonS (Source: Bassem Hassan) and TopBanana *Warren et al., 2018*) reared on a standard *Drosophila* yeast cornmeal medium (7.5 g Agar, 64 g cornmeal, 160 g yeast, 85.5 ml sugar cane syrup in 1 L of water) at 25 °C and 50% relative humidity under a 12/12 light/dark rhythm. Individual flies were placed into the behavioral arenas using a fine brush and were allowed to accommodate for about 10 min before any experiment. The behavioral arenas and flies were cooled on ice for Y-maze experiments before gently loading single flies into the chambers using a fine brush. For experiments with clipped wings, the wings were clipped close to the hinge under $CO_2$ anesthesia, and flies were tested after being allowed to recover within individual vials for at least two days.

## Setups

### Setup 1: Standard Buridan Arena

#### Assay

Two Buridan setups were used: type A uses fluorescent tubes for background illumination (as described in *Colomb et al., 2012*; *Linneweber et al., 2020*). Type B utilizes a panoramical LED matrix (RGB, 384×128 pixels, pixel size 0.9°, 120 Hz frame rate, 5 kHz refresh rate, <1 ms Pixel update latency), which is easily addressable via HDMI for stimulus presentation (see *Figure 1a*). For a detailed description of the assays and tracking procedures, see *Figure 1—figure supplement 1*. Flies were walking on a circular platform (diameter 12 cm) between two opposing black vertical stripes (angular extent 12° horizontal and 90° vertical measured from the center of the platform). A trench of water surrounded the platform to prevent the flies from escaping (type A) or a clear plastic hemispherical dome coated with Sigmacote (Sigma-Aldrich, SL2-25ML) from the inside (type B). Single-fly walking behavior was recorded at 25 °C and 50% relative humidity for 15 min (450×450, 8-bit grayscale, 20 Hz, AVI, Mjpeg compression, 90% quality). Before changing an experimental condition, flies were placed back into single vials for recovery.

#### Automated tracking procedure

Data was analyzed either in real-time using previously described code (type A, see *Colomb et al., 2012*) or offline using a custom-written MATLAB code (type B, available for download from https://github.com/LinneweberLab/Mathejczyk_2024_eLife_Individuality [copy archived at *LinneweberLab, 2026*], see *Figure 1—figure supplement 1c*), respectively. Thresholded video frames and overlaying trajectories are displayed continuously in a separate window to allow the experimenter to assess tracking quality. The MATLAB code works as follows: the circular region of interest (ROI) is automatically detected based on adaptive thresholding and subsequent filtering of the thresholded ROI based on area. Once the ROI is detected, a background image is generated by calculating the maximum intensity between a recorded video's first and last frames. All flies that moved between those frames will not appear in this background-only image. During tracking, the background image is subtracted from each consecutive frame to simplify the thresholding of the fly. For each frame, the centroid of these thresholded pixels is stored. Should more than one object be detected within a ROI (e.g. through pixel noise), the x-/y-coordinates of the object closest to the fly's position in the previous frame are stored. If a fly is lost, the tracker keeps the last known position of the fly until it is detected again.

#### Data output

Once tracking is completed, the raw tracking results are converted to real-world units (mm) and saved into a .mat-file. Before further analysis, minimal movements are filtered out by only updating the fly's position if a certain walking threshold (0.8 mm/frame) is detected. For each fly, a figure can be saved, allowing for a quick overview of the walking trajectory, and a variety of behavioral parameters are also saved into a separate. txt file to simplify further data analysis. For a detailed explanation of calculated

parameters, see *Supplementary file 1*. For an in-depth analysis of Buridan data, sophisticated data analysis routines describing many behavioral parameters were previously developed (*Colomb et al., 2012*). To simplify data integration and compatibility, the raw trajectories can also be saved to a pair of .TXT and .DAT files, which can later be read and analyzed by previously available software (*Colomb et al., 2012*).

## Setup 2: IndyTrax multi-arena platform

### Assay

To allow for high-throughput behavioral quantification under various tightly controlled environmental contexts, we designed the novel assay system IndyTrax. In short, single flies are transferred to an array of small behavioral arenas. They are filmed from below against near-infrared (850 nm + Lee filters 87 C IR-pass filter on camera objective) LED illumination (*Figure 3*). Compared to published Buridan assays (*Colomb et al., 2012*), this allows tracking flies in the absence of visible light, but also has the advantage that flies cannot see the camera lens which might otherwise appear as a high-contrast object from within the arena. The assay was designed to allow for the tracking of flies in various possible arena shapes and numbers. We created three different 3D-printable arena types (*Figure 3—figure supplement 1e–g*), which can be downloaded from https://github.com/LinneweberLab/Mathejczyk_2024_eLife_Individuality (copy archived at *LinneweberLab, 2026*). For individuality analysis, we used the multi-Buridan and Y-maze arenas. The setup was built from commercially available and 3D-printed parts. For a part list, see the *Supplementary file 2*. All 3D-printed parts and behavioral assays were printed on a Prusa i3 mk3 printer using black or white PLA filament (Verbatim 55318) and can be downloaded from https://github.com/LinneweberLab/Mathejczyk_2024_eLife_Individuality. Similarly, the MATLAB code for tracking and data analysis can be downloaded from there. For detailed instructions on how to install and run the code, see the supplemental information. With a computer running MATLAB already available, we estimate the building cost for a complete, temperature- and humidity-controlled enclosure, including all behavioral assays, at about 3500€.

### Temperature- and humidity-controlled enclosure

Behavioral arenas are located within a black, climate-controlled metal enclosure. Temperature and humidity within the enclosure can be regulated using two 3D-printed humidifiers and a heating plate (see *Figure 3—figure supplement 1a*). Ambient temperature and humidity can be set at any desired value using this configuration and kept stable over a long time (see *Figure 3—figure supplement 1c*). Since small, enclosed arenas are prone to heat build-up, we placed all temperature- and humidity sensors within the center arena (arena 13) for Buridan experiments and close to the flies in Y-maze experiments. For Buridan experiments, this limits the number of simultaneously trackable arenas to 24 but allows more accurate climate control within the chambers.

### Small-scale Buridan-stimuli arrays

We modified the basic design of previously published behavioral arenas optimized for Buridan object orientation behavior (*Colomb et al., 2012*) by implementing changes in four major areas to allow for high-throughput tracking: First, we reduced the arena diameter to allow for filming an array (5×5) of optically isolated Buridan arenas with only a single camera (*Figure 3*). Second, flies are walking on circular translucent acrylic plates (49 mm diameter) which allows for filming flies from below against infrared back-illumination, hence avoiding problems with dead angles that would arise when filming cylindrical arrays from above. Third, during experiments, flies are contained on their respective acrylic platform by clear hemispherical plastic domes (*Figure 3b*). This removes the necessity of cutting the flies' wings before experiments and saves space by making the water-filled moats of the original design superfluous. Plastic domes were coated with Sigmacote (Sigma-Aldrich, SL2-25ML) from the inside to prevent the flies from sitting on the dome surface. Fourth, to provide a scalable, homogeneously illuminated white background for optical stimulus presentation, the fluorescent tubes of the original design were replaced with electroluminescent (el-)foil wrapped around translucent acrylic cylinders (*Figure 3b*). Such el-foils can be cut to size and have low heat emission. They emit light evenly over the whole surface, minimizing unwanted intensity cues within the arenas. El foils are driven at 2 kHz, thus providing virtually flicker-free white illumination to the flies. For creating the Buridan object orientation stimulus, thin black stripes (12° wide as measured from the arena center) were 3D

printed and glued vertically to the inside of the arena walls 180° apart, thereby creating high-contrast objects against a uniform white background (*Figure 3b*). Individual arenas were fitted into a 5×5 array. Every other arena within the array was rotated by 90° counter-clockwise to increase the robustness of behavioral readouts by canceling out possible directional artifacts created within the enclosure.

Walking behavior was recorded at 50% relative humidity for 15 min (1200×1200 pixel, 8-bit grayscale, 15 Hz, AVI, Mjpeg compression, 90% quality) with either the el-foil-illumination on, in darkness, at 23 °C or 32 °C, respectively. After being tested in the Buridan assay, the flies were put back into single vials overnight to allow for recovery before testing them in the Y-maze assay the next day around the same time.

### Y-maze array

We constructed a modular 3D-printable Y-maze array (144 arenas, 300×300×2 mm) based on a previously published design (*Werkhoven et al., 2019*). The Y-maze plates were placed between two 320×320×3 mm borosilicate glass plates. The bottom side of the upper glass plate was coated three times using Sigmacote (Sigma-Aldrich, SL2-25ML) to prevent the flies from walking on the upper plate (*Figure 3c*). Directly below the arenas and above the upper glass plate, we placed a layer of white paper (Evercopy premium 80 g, 1902C) for light diffusion and to restrict the view of the flies outside of the arenas. For individuality experiments, we covered the outer arenas with black tape and tracked flies only within 100 arenas located in the center. This made keeping track of individual fly identities easier when switching between the different behavioral assays. Videos of walking flies were recorded at 50% relative humidity for 30 min (1200×1200 pixel, 8-bit grayscale, 15 Hz, AVI, Mjpeg compression, 90% quality) under white light illumination, in darkness, at 23 °C or 32 °C, respectively. For white light illumination, we placed a 30×30 cm LED panel (Tween Light 30×30 cm LED panel, 4000 k, 16 W) on top of the arenas and temporarily removed the infrared-pass filter from the camera objective. Y-maze experiments were all done consecutively on the same day (light on 23 °C, light off 23 °C, light on 32 °C, light off 32 °C).

### Automated tracking procedure

We developed a fully automated MATLAB-based tracking code that allows users to adjust parameters like arena number and size and generates output data for all videos within a user-selectable folder. The code assumes only one fly per region of interest. This saves processing time and increases tracking robustness by not having to match multiple fly identities between frames. Before running the code, the experimenter must create a camera calibration file using the MATLAB camera calibration toolbox (see supplemental information). This allows for more accurate tracking results by correcting lens distortions (*Figure 3—figure supplement 1d*). If the camera stays immobile, this procedure must be done only once, usually in only a few minutes. As long as the arenas filmed are arranged in rows and columns, the code can handle virtually any shape or number of arenas, making it ideal for tracking flies across different behavioral assays (*Figure 3—figure supplement 1e–g*). ROIs are detected based on adaptive thresholding and subsequent filtering of thresholded ROI-objects based on the area (*Figure 3—figure supplement 1e–g*, middle). Once the ROIs are detected and sorted, a background-subtracted image is generated, allowing the user to determine a detection threshold, above which all pixels will be interpreted as foreground objects (*Figure 3—figure supplement 1e*, third from left). During tracking, background subtraction and object thresholding is performed for each frame, and for each ROI, the centroid of the thresholded fly is extracted and saved. Thresholded pixels and trajectories are displayed continuously in a separate window to allow the experimenter to assess tracking quality. If more than one object is detected within a ROI (e.g. through pixel noise), the x-/y--coordinates of the object closest to the fly's position in the previous frame is stored. If a fly is lost, the tracker keeps the last known position of the fly until it is detected again. The average tracking speed for 24 ROIs/flies was about 20 hz with MATLAB running on an Intel i5 3 Ghz CPU. For tracking 400 ROIs/flies, the average tracking speed was about 5 hz.

### Data output

Data output is the same as described for our standard Buridan arenas.

## Setup 3: Virtual flight simulator

### Assay

We designed a novel, easy-to-assemble, and easy-to-use virtual flight simulator using affordable 3D-printable and off-the-shelf parts (*Figure 5a–c*) to quantify behavioral responses during flight. For a detailed description of the setup and custom code, see *Figure 5—figure supplement 1*. In short, using a tethering station (for description, see *Mathejczyk and Wernet, 2020*), flies were cooled down for immobilization and glued to a steel pin (0.1 mm diameter, 10 mm length, https://www.entosphinx.cz/) using UV-cured glue (Bondic). Once tethered, single flies were put between two neodymium magnets (magneto-tether; upper magnet: diameter 5 mm, lower magnet: ring, outer diameter 10 mm, inner diameter 5 mm). Once the appropriate distance between the magnet is found (about 20 mm), the magnetic field allows the flies to rotate freely around their yaw axis while keeping the steel pin in place (*Figure 5b*). The tip of the steel pin is inserted into a V-shaped sapphire bearing (1 mm diameter) to reduce friction. For the re-initiation of flight, air puffs can be delivered from below via a membrane pump. A panoramic LED matrix surrounding the magneto-tether was used for Buridan stimulus presentation (RGB, 256×128 pixels, 120 Hz frame rate, 5 kHz refresh rate, pixel size 1.4°, angular extent 360° horizontal × 100° vertical, stripe width: 12°). The LED matrix controller is connected to a computer via HDMI, allowing an easy and flexible stimulus creation and presentation (Novastar mctrl 660 pro, 120 Hz, <1 ms Pixel update latency). Flies were filmed from below under near-infrared illumination (*Figure 5c*), and each fly's heading was tracked in real-time (90 Hz) using a custom-written MATLAB code (available for download from https://github.com/LinneweberLab/Mathejczyk_2024_eLife_Individuality, copy archived at *LinneweberLab, 2026*). Before starting the experiment, each fly was tested for 120 s with an optomotor stimulus (rotating stripe pattern, 120°/s, 40 x CW and 40 x CCW, respectively) to assess the flies' ability to rotate smoothly (*Figure 5d*, left). After this, flies were presented with the Buridan stimulus with varying contrast (100%, 50%, 100% inverted; stripe width 30°), and their heading was tracked for 3×4 min. Flies were then carefully separated from the steel pin and transferred into single vials for overnight recovery before being re-tethered and tested the following day around the same time. For flight vs. walking experiments, flies were tested for 10 min while flying under a 100% contrast Buridan stimulus (stripe width 12° from a fly's perspective) before being gently separated from the steel pin and being transferred into the LED Buridan assay (Type B), where they were tested for 15 min under a 100% contrast Buridan stimulus (stripe width 12° from a fly's perspective).

### Automated tracking procedure

The code for tracking a fly's heading in 360° can be downloaded from https://github.com/LinneweberLab/Mathejczyk_2024_eLife_Individuality (copy archived at *LinneweberLab, 2026*). Flight tracking is done in real-time at 90fps (*Figure 5—figure supplement 1c*). A timestamp is created for each incoming camera frame, and two different thresholded images are computed with different Gaussian filter settings. A strong filter setting blurs out the legs of the fly and allows for thresholding only the fly's body and calculating a heading angle between 0° and 180° through ellipse fitting. By subtracting the thresholded body pixels from a weaker filtered version of the same frame, the limbs and wings (visible when the fly stops flying) can also be tracked separately if the experimenter wants. To calculate the heading between 0° and 360°, the centroid of thresholded body pixels is set to (0,0) within a fictive coordinate system (see *Figure 5—figure supplement 1c*), and the body axis is aligned with the x-axis. The head region is assigned based on the sum of pixels within the 4-axis quadrants (smallest area of either I+IV or II +III, respectively). Once the heading is calculated, it is stored together with the original video frame for additional offline data analysis, if desired. When a fly stops flying (wings get detected in the video), a command is sent to a connected Arduino, which controls a relay to switch on a membrane pump and provide an air puff to reinitiate flight. During tracking, the thresholded video is displayed together with the computed heading to allow the experimenter to assess tracking quality.

### Data output

After tracking, raw heading data, a graphical overview image, and a table containing computed behavioral parameters are saved automatically for each fly which can be used for further data analysis. For a detailed explanation of all calculated parameters, see *Supplementary file 1*.

## Data and statistical analysis for all experiments

All data analysis was performed with MATLAB. For angular calculations, we used the algorithm presented in *Berens, 2009*. After a test for normality (Shapiro-Wilk test), behavioral data between groups was compared using the paired T-test. For correlating individual data, we calculated Pearson correlation coefficients. For walking data, an arena's center region (edge-corrected area) was defined as the circular area around the center with a radius of 80% of the arena radius. The maximum color value (red) for generating heatmaps was set to the 95%-quantile of the count distribution to increase data visibility. Flies that did not move during the experiment were excluded from the analysis.

## Generalized linear model (GLM)

To summarize and generalize the findings from all experiments in this study (using individual rank differences upon changing one or more environmental parameters), a GLM was trained using the environmental parameters as predictors (0 when environmental parameter was not changed and 1 if it was changed for any given experimental group) and the resulting individual rank differences as the response variable. Since the five environmental parameters are not fully statistically independent (e.g. arena and vision were altered only in walking experiments and not in flight), we employed ridge regression to mitigate collinearity among these predictors and to control for overfitting. In contrast to stepwise or LASSO regression, ridge regression allowed us to keep all environmental parameters as part of the model. A range of regularization parameters ($\lambda$) was tested using fivefold cross-validation over a log-spaced grid ($10^{-6}$ - $10^{6}$) to identify the optimal value that minimized the mean squared error (MSE). Z-scoring was applied to the predictor variables prior to model fitting for data normalization. To estimate coefficient variability and assess significance, bootstrapping was applied. For each of the 1000 bootstrap iterations, predictors and response data were resampled with replacement, and the ridge regression model was refit using the optimal $\lambda$. Coefficients and intercept values from each bootstrap sample were stored, and their distributions were used to calculate the mean coefficients, standard errors (SEs), and p-values for coefficients based on t-statistics derived from bootstrapped SEs.

## Hierarchical linear mixed model

We collected behavioral data from all tested walking and flight assays and summarized the five previously quantified behavioral traits under different environmental contexts for walking and flight behaviors, respectively. For each trait, we fitted a hierarchical linear mixed-effects model in Matlab (using the fit lme function) with environmental context as a fixed effect and fly identity (ID) as a random intercept. Fixed effects were included to account for systematic differences due to the different environmental contexts whenever multiple context levels were tested for that trait. This controls for behavioral population-level shifts caused by changing the day, temperature, visual conditions, or the arena shape. Such a model partitions variance into within-fly and between-fly components. We computed the intraclass correlation coefficient (ICC) from each model as the between-fly variance divided by total variance. ICC, therefore, quantified repeatability across environmental contexts. We estimated 95% confidence intervals (CIs) for ICC using non-parametric bootstrapping (flies were resampled with replacement across contexts, and models were refitted 1000 times).

## Acknowledgements

The authors thank the Bloomington stock center and Eugenia Chiappe for fly stocks and reagents. This work was supported by the Deutsche Forschungsgemeinschaft (DFG) through the DFG research unit 5289 RobustCircuit (GAL, MFW), through grants LI 2640/1- 1, LI 2640/2-1 (GAL), WE 5761/4-1 (MFW), SPP 2205 (MFW), through AFOSR grants FA9550-19-1-7005 / FA 8655-23-1-7049 (MFW), and with support from the Fachbereich Biologie, Chemie & Pharmazie of the Freie Universität Berlin, as well as the Division of Neurobiology at Freie Universität Berlin. We acknowledge support by the Open Access Publication Fund of Freie Universität Berlin. We thank members of the Linneweber and Wernet labs, Randolf Menzel, and Robin Hiesinger for helpful discussions.

## Additional information

### Funding

| Funder | Grant reference number | Author |
|---|---|---|
| Deutsche Forschungsgemeinschaft | LI 2640/1-1 | Gerit A Linneweber |
| Deutsche Forschungsgemeinschaft | FOR5289 LI 2640/2-1 | Gerit A Linneweber |
| Deutsche Forschungsgemeinschaft | FOR5289 WE 5761/4-1 | Mathias F Wernet |

The funders had no role in study design, data collection and interpretation, or the decision to submit the work for publication.

### Author contributions

Thomas F Mathejczyk, Conceptualization, Data curation, Formal analysis, Visualization; Cara Knief, Muhammad A Haidar, Tydings McClary, Data curation; Florian Freitag, Formal analysis; Mathias F Wernet, Supervision, Funding acquisition, Writing – review and editing; Gerit A Linneweber, Conceptualization, Resources, Supervision, Funding acquisition, Writing – original draft, Project administration, Writing – review and editing

### Author ORCIDs

Mathias F Wernet ⓘ https://orcid.org/0000-0001-5233-2654
Gerit A Linneweber ⓘ https://orcid.org/0000-0001-8393-6426

Joint Public Review: https://doi.org/10.7554/eLife.98171.5.sa1
Author response https://doi.org/10.7554/eLife.98171.5.sa2

## Additional files

### Supplementary files

Supplementary file 1. Description of automatically computed output parameters.

Supplementary file 2. Part list for setup replication.

Supplementary file 3. Fixed-effects summary from hierarchical linear mixed model analysis for all behavioral traits. For each behavioral trait, the model included environmental context variables as fixed effects and fly identity (ID) as a random intercept. Name: fixed-effect terms. Estimate: estimated effect size on the respective trait (in trait units). SE: standard error of estimate. DF: degrees of freedom used for testing significance. tStat: t-statistic. p-value: p-value. Lower/upper: bound of the 95% confidence interval for the effect estimate. Trait: behavioral trait.

MDAR checklist

### Data availability

All experimental data are available here: https://doi.org/10.5281/zenodo.18342691. The code for the analyses presented in this paper is openly accessible at https://github.com/LinneweberLab/Mathejczyk_2024_eLife_Individuality (copy archived at *LinneweberLab, 2026*).

The following dataset was generated:

| Author(s) | Year | Dataset title | Dataset URL | Database and Identifier |
|---|---|---|---|---|
| Thomas FM, Cara K, Muhammad AH, Florian F, Tydings M, Mathias W, Gerit L | 2026 | Individuality across environmental context in *Drosophila melanogaster* | https://doi.org/10.5281/zenodo.18342691 | Zenodo, 10.5281/zenodo.18342691 |

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
