## [Editor Report · eLife Assessment]

There is a growing interest in understanding the individuality of animal behaviours. In this **important** article, the authors build and use an impressive array of high throughput phenotyping paradigms to examine the 'stability' (consistency) of behavioural characteristics in a range of contexts and over time. The results show that certain behaviours are individualistic and persist robustly across external stimuli while others are less robust to these changing parameters. The data supporting their findings is extensive and **convincing**. At the same time, the main analyses focus on a selected subset of the many behavioural metrics recorded, so a large fraction of the acquired data remains only lightly explored; by making these additional data available, the authors provide an invaluable resource for future work to apply alternative analytical frameworks and further mine this rich dataset.

---

## [Referee Report · Joint Public Review]

Summary:

The authors state the study's goal clearly: "The goal of our study was to understand to what extent animal individuality is influenced by situational changes in the environment, i.e., how much of an animal's individuality remains after one or more environmental features change." They use visually guided behavioral features to examine the extent of correlation over time and in a variety of contexts. They develop new behavioral instrumentation and software to measure behavior in Buridan's paradigm (and variations thereof), the Y-maze, and a flight simulator. Using these assays, they examine the correlations between conditions for a panel of locomotion parameters. They propose that inter-assay correlations will determine the persistence of locomotion individuality.

Comments from the editors on the latest version:

In the latest communication, the authors were asked to (i) justify their selection of metrics (i.e. why these specific five behavioural metrics were chosen from the many recorded), (ii) discuss the variation in ICCs, and (iii) in light of this variation and the reliance on a few selected behavioural parameters, tone down the general claim so as not to overstate that individuality persists across all behaviours.

We note that the justification for choosing the five metrics and the discussion of ICC variation are purely qualitative, and, despite the edits, the manuscript continues to frame individual behaviours as broadly stable.

---

## [Author Response]

The following is the authors’ response to the previous reviews

We appreciate the authors' efforts in addressing the concerns raised, particularly including a variance partitioning approach to analyse their data. Detailed feedback on the revised manuscript are below and we include a brief list of comments that we think the authors could address in the text:

(1) Justify metric selection - Could you please include in the text and explanation for why only five behavioural metrics were highlighted out of the many you calculated?

We have added explanations throughout the manuscript clarifying the rationale for selecting these behavioral parameters, including in lines 467ff. and 531ff. In short, the five highlighted metrics were chosen because they capture key aspects of the behavioral repertoire and, importantly, can be consistently measured across all experimental conditions. Other parameters were excluded as they were only applicable under specific contexts and thus not suitable for cross-condition comparisons.

(2) Discuss ICC variation - We note that there is variation among the ICC scores for the different metrics you've studied. While this is expected, we ask that you acknowledge in the text that some traits show high repeatability and others low, and reflect this variation in the conclusions.

We have added an additional paragraph in the Discussion (lines 743ff.) addressing the variation in ICC values among behavioral traits. This new section highlights that some metrics show high repeatability while others exhibit lower consistency, and we discuss how this heterogeneity informs our conclusions about individual behavioral stability across contexts.

(3) Tone down general claims - Because of the above point, we recommend that you avoid overstating that individuality persists across all behaviours. Please clarify this in the Abstract and main text that it applies to some traits more than others.

We carefully reviewed the entire manuscript and revised the phrasing wherever necessary to avoid overgeneralization. Statements about individuality have been adjusted to clarify that consistent individuality can be measured in some behavioral traits more strongly than to others, both in the Abstract and throughout the main text.

**Public Reviews:**

**Reviewer #1 (Public review):**
Summary:The authors state the study's goal clearly: "The goal of our study was to understand to what extent animal individuality is influenced by situational changes in the environment, i.e., how much of an animal's individuality remains after one or more environmental features change." They use visually guided behavioral features to examine the extent of correlation over time and in a variety of contexts. They develop new behavioral instrumentation and software to measure behavior in Buridan's paradigm (and variations thereof), the Y-maze, and a flight simulator. Using these assays, they examine the correlations between conditions for a panel of locomotion parameters. They propose that inter-assay correlations will determine the persistence of locomotion individuality.Strengths:The OED defines individuality as "the sum of the attributes which distinguish a person or thing from others of the same kind," a definition mirrored by other dictionaries and the scientific literature on the topic. The concept of behavioral individuality can be characterized as: (1) a large set of behavioral attributes, (2) with inter-individual variability, that are (3) stable over time. A previous study examined walking parameters in Buridan's paradigm, finding that several parameters were variable between individuals, and that these showed stability over separate days and up to 4 weeks (DOI: 10.1126/science.aaw718). The present study replicates some of those findings, and extends the experiments from temporal stability to examining correlation of locomotion features betweendifferent contexts.The major strength of the study is using a range of different behavioral assays to examine the correlations of several different behavior parameters. It shows clearly that the inter-individual variability of some parameters is at least partially preserved between some contexts, and not preserved between others. The development of highthroughput behavior assays and sharing the information on how to make the assays is a commendable contribution.Weaknesses:The definition of individuality considers a comprehensive or large set of attributes, but the authors consider only a handful. In Supplemental Fig. S8, the authors show a large correlation matrix of many behavioral parameters, but these are illegible and are only mentioned briefly in Results. Why were five or so parameters selected from the full set? How were these selected? Do the correlation trends hold true across all parameters? For assays in which only a subset of parameters can be directly compared, were all of these included in the analysis, or only a subset?The correlation analysis is used to establish stability between assays. For temporal retesting, "stability" is certainly the appropriate word, but between contexts it implies that there could be 'instability'. Rather, instead of the 'instability' of a single brain process, a different behavior in a different context could arise from engaging largely (or entirely?) distinct context-dependent internal processes, and have nothing to do with process stability per se. For inter-context similarities, perhaps a better word would be "consistency".The parameters are considered one-by-one, not in aggregate. This focuses on the stability/consistency of the variability of a single parameter at a time, rather than holistic individuality. It would appear that an appropriate measure of individuality stability (or individuality consistency) that accounts for the high-dimensional nature of individuality would somehow summarize correlations across all parameters. Why was a multivariate approach (e.g. multiple regression/correlation) not used? Treating the data with a multivariate or averaged approach would allow the authors to directly address 'individuality stability', along with the analyses of single-parameter variability stability.The correlation coefficients are sometimes quite low, though highly significant, and are deemed to indicate stability. For example, in Figure 4C top left, the % of time walked at 23°C and 32°C are correlated by 0.263, which corresponds to an R2 of 0.069 i.e. just 7% of the 32°C variance is predictable by the 23°C variance. Is it fair to say that 7% determination indicates parameter stability? Another example: "Vector strength was the most correlated attention parameter... correlations ranged... to -0.197," which implies that 96% (1 - R2) of Y-maze variance is not predicted by Buridan variance. At what level does an r value not represent stability?The authors describe a dissociation between inter-group differences and interindividual variation stability, i.e. sometimes large mean differences between contexts, but significant correlation between individual test and retest data. Given that correlation is sensitive to slope, this might be expected to underestimate the variability stability (or consistency). Is there a way to adjust for the group differences before examining correlation? For example, would it be possible to transform the values to ingroup ranks prior to correlation analysis?What is gained by classifying the five parameters into exploration, attention, and anxiety? To what extent have these classifications been validated, both in general, and with regard to these specific parameters? Is increased walking speed at higher temperature necessarily due to increased 'explorative' nature, or could it be attributed to increased metabolism, dehydration stress, or a heat-pain response? To what extent are these categories subjective?The legends are quite brief and do not link to descriptions of specific experiments. For example, Figure 4a depicts a graphical overview of the procedure, but I could not find a detailed description of this experiment's protocol.Using the current single-correlation analysis approach, the aims would benefit from rewording to appropriately address single-parameter variability stability/consistency (as distinct from holistic individuality). Alternatively, the analysis could be adjusted to address the multivariate nature of individuality, so that the claims and the analysis are in concordance with each other.The study presents a bounty of new technology to study visually guided behaviors. The Github link to the software was not available. To verify successful transfer or openhardware and open-software, a report would demonstrate transfer by collaboration with one or more other laboratories, which the present manuscript does not appear to do. Nevertheless, making the technology available to readers is commendable.The study discusses a number of interesting, stimulating ideas about inter-individual variability, and presents intriguing data that speaks to those ideas, albeit with the issues outlined above.While the current work does not present any mechanistic analysis of inter-individual variability, the implementation of high-throughput assays sets up the field to more systematically investigate fly visual behaviors, their variability, and their underlying mechanisms.Comments on revisions:While the incorporation of a hierarchical mixed model (HMM) appears to represent an improvement over their prior single-parameter correlation approach, it's not clear to me that this is a multivariate analysis. They write that "For each trait, we fitted a hierarchical linear mixed-effects model in Matlab (using the fit lme function) with environmental context as a fixed effect and fly identity (ID) as a random intercept... We computed the intraclass correlation coefficient (ICC) from each model as the betweenfly variance divided by total variance. ICC, therefore, quantified repeatability across environmental contexts."Does this indicate that HMM was used in a univariate approach? Can an analysis of only five metrics of several dozen total metrics be characterized as 'holistic'?Within Figure 10a, some of the metrics show high ICC scores, but others do not. This suggests that the authors are overstating the overall persistence and/or consistency of behavioral individuality. It is clear from Figure S8 that a large number of metrics were calculated for each fly, but it remains unclear, at least to me, why the five metrics in Figure 10a are justified for selection. One is left wondering how rare or common is the 0.6 repeatability of % time walked among all the other behavioral metrics. It appears that a holistic analysis of this large data set remains impossible.

We thank the reviewer for the careful and thoughtful assessment of our work.

We have added an additional paragraph in the Discussion (lines 743ff.) explicitly addressing the variation in ICC values among behavioral traits. This section emphasizes that while some metrics show high repeatability, others exhibit lower consistency, and we discuss how this heterogeneity informs our conclusions regarding individual behavioral stability across contexts.

Regarding the reviewer’s concern about the analytical approach, we would like to clarify that the hierarchical linear mixed model (LMM) was applied in a univariate framework—each behavioral metric was analyzed separately to estimate its individual ICC value. This approach allows us to quantify repeatability for each trait across environmental contexts while accounting for individual identity as a random effect. Although this is not a multivariate model in the strict sense, it represents an improvement over the prior pairwise correlation approach because it explicitly partitions within- and between-individual variance.

As for the selection of behavioral metrics, the five parameters highlighted (% time walked, walking speed, vector strength, angular velocity, and centrophobicity) were chosen because they represent key, biologically interpretable dimensions of locomotor and spatial behavior and, importantly, could be measured reliably across all tested conditions. Several other parameters that we routinely analyze (e.g., Linneweber et al., 2020) could not be calculated in all contexts—for instance, under darkness or when visual cues were absent—and therefore were excluded to maintain consistency across assays.

We agree that a truly holistic multivariate comparison across all extracted parameters would be valuable; however, given the contextual limitations of some metrics, such an analysis was not feasible in the present framework. We have clarified these points in the revised manuscript to avoid potential misunderstandings.

The authors write: "...fly individuality persists across different contexts, and individual differences shape behavior across variable environments, thereby making the underlying developmental and functional mechanisms amenable to genetic dissection." However, presumably the various behavioral features (and their variability) are governed by different brain regions, so some metrics (high ICC) would be amenable to the genetic dissection of individuality/variability, while others (low ICC) would not. It would be useful to know which are which, to define which behavioral domains express individuality, and could be targets for genetic analysis, and which do not. At the very least, the Abstract might like to acknowledge that inter-context consistency is not a major property of all or most behavioral metrics.

We thank the reviewer for this helpful comment and agree that not all behavioral traits exhibit the same degree of inter-context consistency. We have clarified this point in the revised Abstract and ensured that it is also reflected in the main text. The Abstract now reads:

“We find that individuality is highly context-dependent, but even under the most extreme environmental alterations tested, consistency of behavioral individuality always persisted in at least one of the traits. Furthermore, our quantification reveals a hierarchical order of environmental features influencing individuality. We confirmed this hierarchy using a generalized linear model and a hierarchical linear mixed model. In summary, our work demonstrates that, similar to humans, fly individuality persists across different contexts (albeit worse than across time), and individual differences shape behavior across variable environments. The presence of consistency across situations in flies makes the underlying developmental and functional mechanisms amenable to genetic dissection.”

This revision clarifies that individuality is not uniformly expressed across all behavioral metrics, but rather in a subset of traits with higher repeatability, which are the most promising targets for future genetic analyses.

I hold that inter-trial repeatability should rightly be called "stability" while inter-context repeatability should be called "consistency". In the current manuscript, "consistency" is used throughout the manuscript, except for the new edits, which use "stability". If the authors are going to use both terms, it would be preferable if they could explain precisely how they define and use these terms.

We thank the reviewer for drawing attention to this inconsistency in terminology. We apologize for the oversight and have corrected it throughout the manuscript to ensure uniform usage.

**Reviewer #2 (Public review):**
Summary:The authors repeated measured the behavior of individual flies across several environmental situations in custom-made behavioral phenotyping rigs.Strengths:The study uses several different behavioral phenotyping devices to quantify individual behavior in a number of different situations and over time. It seems to be a very impressive amount of data. The authors also make all their behavioral phenotyping rig design and tracking software available, which I think is great and I'm sure other folks will be interested in using and adapting to their own needs.Weaknesses/Limitations:I think an important limitation is that while the authors measured the flies under different environmental scenarios (i.e. with different lighting, temperature) they didn't really alter the "context" of the environment. At least within behavioral ecology, context would refer to the potential functionality of the expressed behaviors so for example, an anti-predator context, or a mating context, or foraging. Here, the authors seem to really just be measuring aspects of locomotion under benign (relatively low risk perception) contexts. This is not a flaw of the study, but rather a limitation to how strongly the authors can really say that this demonstrates that individuality is generalized across many different contexts. It's quite possible that rank-order of locomotor (or other) behaviors may shift when the flies are in a mating or risky context.I think the authors are missing an opportunity to use much more robust statistical methods. It appears as though the authors used pearson correlations across time/situations to estimate individual variation; however far more sophisticated and elegant methods exist. The problem is that pearson correlation coefficients can be anticonservative and additionally, the authors have thus had to perform many many tests to correlate behaviors across the different trials/scenarios. I don't see any evidence that the authors are controlling for multiple testing which I think would also help. Alternatively, though, the paper would be a lot stronger, and my guess is, much more streamlined if the authors employ hierarchical mixed models to analyse these data, which are the standard analytical tools in the study of individual behavioral variation. In this way, the authors could partition the behavioral variance into its among- and withinindividual components and quantify repeatability of different behaviors across trials/scenarios simultaneously. This would remove the need to estimate 3 different correlations for day 1 & day 2, day 1 & 3, day 2 & 3 (or stripe 0 & stripe 1, etc) and instead just report a single repeatability for e.g. the time spent walking among the different strip patterns (eg. figure 3). Additionally, the authors could then use multivariate models where the response variables are all the behaviors combined and the authors could estimate the among-individual covariance in these behaviors. I see that the authors state they include generalized linear mixed models in their updated MS, but I struggled a bit to understand exactly how these models were fit? What exactly was the response? what exactly were the predictors (I just don't understand what Line404 means "a GLM was trained using the environmental parameters as predictors (0 when the parameter was not change, 1 if it was) and the resulting individual rank differences as the response"). So were different models run for each scenario? for different behaviors? Across scenarios? what exactly? I just harp on this because I'm actually really interested in these data and think that updating these methods can really help clarify the results and make the main messages much clearer!I appreciate that the authors now included their sample sizes in the main body of text (as opposed to the supplement) but I think that it would still help if the authors included a brief overview of their design at the start of the methods. It is still unclear to me how many rigs each individual fly was run through? Were the same individuals measured in multiple different rigs/scenarios? Or just one?I really think a variance partitioning modeling framework could certainly improve their statistical inference and likely highlight some other cool patterns as these methods could better estimate stability and covariance in individual intercepts (and potentially slopes) across time and situation. I also genuinely think that this will improve the impact and reach of this paper as they'll be using methods that are standard in the study of individual behavioral variation
**Recommendations for the authors:**

**Reviewer #2 (Recommendations for the authors):**
I am delighted to see the authors have included hierarchical models in their analysis. I really think this strengthens the paper and their conclusions while simultaneously making it more accessible to folks that typically use these types of methods to investigate these patterns of individual behavior. It's also cool, and completely jives with my own experience measuring individual behavior in that the activity metrics show the highest repeatability compared to the more flexible behaviors (such as "exploration"). I think it's quite striking and interesting to see such moderate repeatability estimates in these behaviors across what could be very different environmental scenarios. I think this is a very strong and meaty paper with a lot of information to digest producinghowever a very elegant and convincing take-home message: individuals are unique in their behavior even across very different environments.

We sincerely thank the reviewer for the positive and encouraging feedback, as well as for their valuable input throughout the review process. We are very pleased that the inclusion of hierarchical models and the resulting interpretations resonated with the reviewer’s own experience and perspective.